

# DPWSS: differentially private working set selection for training support vector machines

Zhenlong Sun[1,2], Jing Yang[1], Xiaoye Li[2] and Jianpei Zhang[1]

[1] College of Computer Science and Technology, Harbin Engineering University, Harbin, Heilongjiang, China
[2] College of Computer and Control Engineering, Qiqihar University, Qiqihar, Heilongjiang, China

## ABSTRACT

Support vector machine (SVM) is a robust machine learning method and is widely used in classification. However, the traditional SVM training methods may reveal personal privacy when the training data contains sensitive information. In the training process of SVMs, working set selection is a vital step for the sequential minimal optimization-type decomposition methods. To avoid complex sensitivity analysis and the influence of high-dimensional data on the noise of the existing SVM classifiers with privacy protection, we propose a new differentially private working set selection algorithm (DPWSS) in this paper, which utilizes the exponential mechanism to privately select working sets. We theoretically prove that the proposed algorithm satisfies differential privacy. The extended experiments show that the DPWSS algorithm achieves classification capability almost the same as the original non-privacy SVM under different parameters. The errors of optimized objective value between the two algorithms are nearly less than two, meanwhile, the DPWSS algorithm has a higher execution efficiency than the original non-privacy SVM by comparing iterations on different datasets. To the best of our knowledge, DPWSS is the first private working set selection algorithm based on differential privacy.

## INTRODUCTION

In recent years, with the rapid development of artificial intelligence, cloud computing, and big data technologies, data sharing and analysis are becoming easier and more practical. A large amount of individual information is stored in electronic databases, such as economic records, medical records, web search records, and social network data, which poses a great threat to personal privacy. Support vector machine (SVM) is one of the most widely used and robust machine learning methods for classification. *Boser, Guyon & Vapnik (1992)* proposed the earliest SVM classification idea by maximizing the margin between the training patterns and the decision boundary. *Cortes & Vapnik (1995)* solved the classification problem of non separable training data through non linearly mapping them to a very high dimension feature space. *Vapnik & Vapnik (1998)* considered three different kernels to construct learning machines with different types of nonlinear decision surfaces in the input space. *Burges (1998)* gave an overview on linear SVMs and kernel

Corresponding author
Jing Yang, yangjing@hrbeu.edu.cn

SVMs with numerous examples for pattern recognition. *Chang & Lin (2007)* developed a popular library LIBSVM for SVMs and presented all the implementation details. A SVM trains a classification model by solving an optimization problem and requires only as few as a dozen examples for training. However, when the training data sets contain sensitive information, directly releasing the SVM classification model may reveal personal privacy.

Generally speaking, training SVMs is to solve a large optimization problem of quadratic programming (QP). Sequential minimal optimization (SMO) (*Platt, 1999*) is currently a commonly used decomposition method for training SVMs by solving the smallest QP optimization problem, and only needs two elements in every iteration. *Keerthi, Shevade & Bhattacharyya (2001)* employed two threshold parameters to derive modifications of SMO and it performed significantly faster than the original SMO algorithm. In all kinds of SMO-type decomposition methods, working set selection (WSS) is an important step. Different WSS algorithms determine the convergence efficiency of the SVM training process. *Zuo, Yi & Lv (2010)* proposed an improved WSS and a simplified minimization step for the SMO-type decomposition method.

Differential privacy (DP) was proposed by a series of work of *Dwork (2006)* from 2006, which has been becoming an accepted standard for privacy protection in sensitive data analysis. DP ensures that adding or removing a single item does not affect the analysis outcome too much, and the privacy level is quantified by a privacy budget $\varepsilon$. DP is realized by introducing randomness or uncertainty. According to the difference of data types, it mainly includes Laplace mechanism (*McSherry & Talwar, 2007*), Gaussian mechanism, and exponential mechanism (*Dwork et al., 2006*). Among them, the Laplace mechanism and Gaussian mechanism are mostly used for numerical data, while the exponential mechanism is used for non-numerical data.

In this paper, we studied the privacy leakage problem of the traditional SVM training methods. There are some shortcomings in the existing SVM classifiers with privacy protection, such as the low classification accuracy, the requirements on the differentiability of the objective function, the complex sensitivity analysis, and the influence of high-dimensional data on noise. We give a solution by introducing randomness in the training process of SVMs to privately release the classification model. The main contributions in this paper conclude as follows:

- We propose an improved WSS method for training SVMs and design a simple scoring function for the exponential mechanism, in which the sensitivity is easy to analyze.
- We propose a new differentially private working set selection algorithm (DPWSS) based on the exponential mechanism, which is achieved by privately selecting the working set in every iteration.
- To improve the utilization of the privacy budget, every violating pair is selected only once during the entire training process.
- We analyze theoretically that the DPWSS algorithm satisfies the requirement of DP, and evaluate the classification capability, algorithm stability, and execution efficiency of the

DPWSS algorithm *vs* the original non-privacy SVM algorithm through extended experiments.

The rest of this paper is organized as follows. Section "Related Work" discusses related work. Section "Preliminaries" introduces the background knowledge of SVMs, WSS, and DP. Section "DPWSS Algorithm" proposes a novel DPWSS algorithm. Section "Experiments" gives the experimental evaluation of the performance of DPWSS. Lastly, Section "Conclusions" concludes the research work.

## RELATED WORK

In this section, we briefly review some work related to privacy-preserving SVMs. *Mangasarian, Wild & Fung (2008)* considered the classification problem of sharing private data by separating agents and proposed using random kernels for vertically partitioned data. *Lin & Chen (2011)* pointed out an inherent privacy violation problem of support vectors and proposed a privacy-preserving SVM classifier, PPSVC, which replaces the Gaussian kernel with an approximate decision function. In these two methods, the degree of privacy protection cannot be proved as the private SVMs based on DP.

As DP is becoming an accepted standard for private data analysis, some SVM classification models based on DP have produced in the recent two decades. Chaudhuri et al. proposed two popular perturbation-based techniques output perturbation and objective perturbation (*Chaudhuri & Monteleoni, 2009*; *Chaudhuri, Monteleoni & Sarwate, 2011*). Output perturbation introduces randomness into the weight vector *w* after the optimization process, and the randomness scale is determined by the sensitivity of *w*. On the contrary, objective perturbation introduces randomness into the objective function before the optimization, and the randomness scale is independent of the sensitivity of *w*. However, the sensitivity of the two perturbation-based techniques is difficult to analyze (*Liu, Li & Li, 2017*) and the objective perturbation requires the loss function satisfying certain convexity and differentiability criteria. *Rubinstein et al. (2012)* proposed a private kernel SVM algorithm PrivateSVM for convex loss functions with Fourier transformation and output perturbation to release the private SVM classification model. However, the classification model is valid only for the translation-invariant kernels. To alleviate too much noise in the final outputs, *Li et al. (2014)* developed a hybrid private SVM model that uses a small portion of public data to calculate the Fourier transformation. However, public data is hard to obtain in the modern private world. *Zhang, Hao & Wang (2019)* constructed a novel private SVM classifier by dual variable perturbation, which adds Laplace noise to the corresponding dual variables according to the ratio of errors.

Different from those kinds of perturbation-based techniques mentioned above, which introduce randomness into the output result or objective function, the DPWSS algorithm introduces randomness during the process of WSS. Therefore, it avoids complex sensitivity analysis and the influence of high-dimensional data on noise, meanwhile improves the performance of the classification model to some extent.

**Table 1 Notations.**

| Symbol | Description |
|--------|-------------|
| $U$ | Universe |
| $D \subset U$ | Dataset to be trained |
| $D'$ | Neighbor dataset of $D$ |
| $x^i \in R^d$ | Train instance |
| $y^i \in \{1, -1\}$ | Label of train instance |
| $\alpha$ | Dual variable |
| $e$ | Vector of all ones |
| $C$ | Upper bound of all variables |
| $K$ | Kernel function |
| $Q$ | Symmetric matrix of kernel function |
| $B$ | Working set |
| $\tau$ | A small positive number |
| $\sigma$ | Constant-factor |
| $M$ | Mechanism |
| $Lap(\lambda)$ | Laplace distribution with mean 0 and scale factor $\lambda$ |
| $\varepsilon$ | Privacy budget |
| $f$ | Query function |
| $q(D, r)$ | Score function |
| $\Delta f, \Delta q$ | Sensitivity of function |
| $TP$ | True positive |
| $TN$ | True negative |
| $FP$ | False positive |
| $FN$ | False negative |

## PRELIMINARIES

In this section, we introduce some background knowledge of SVM, WSS, and DP. Table 1 summarizes the notations in the following sections.

### Support vector machines

The SVM is an efficient classification method in machine learning that originates from structural risk minimization (*Vapnik & Vapnik, 1998*). It finds an optimal separating hyperplane with the maximal margin to train a classification model. Given training instances $x_i \in R^n$ and labels $y_i \in \{1, -1\}$, the main task for training a SVM is to solve the QP optimization problem as follows (*Fan, Chen & Lin, 2005*):

$$
\min_{\alpha} f(\alpha) = \frac{1}{2}\alpha^T Q \alpha - e^T \alpha
$$
$$
\text{Subject to} \quad 0 \leq \alpha_i \leq C, i = 1, \ldots, l
$$
$$
y^T \alpha = 0
$$
(1)

where $Q$ is a symmetric matrix with $Q_{ij} = y_i y_j K(x_i, x_j)$, and $K$ is the kernel function, $e$ is a vector with all 1's, $C$ is the upper bound of vector $\alpha$.

## Working set selection

Generally, the QP problem is hard to solve in the training process of the SVMs. When the optimization methods handle the large matrix $Q$, the whole vector $\alpha$ will be updated repeatedly in the iterative process. Nevertheless, the decomposition methods only update a subset of vector $\alpha$ in every iteration to solve the challenge and change from one iteration to another. The subset is called the working set. The method for determining the working set is called WSS, which originally derives from the optimality conditions of Karush-Kuhn-Tucker (KKT). Furthermore, SMO-type decomposition methods restrict the working set to only two elements (*Platt, 1999*). A pair of elements that violate the KKT optimality conditions are called "violating pair" (*Keerthi, Shevade & Bhattacharyya, 2001*).

**Definition 1** (Violating pair (*Keerthi, Shevade & Bhattacharyya, 2001*; *Fan, Chen & Lin, 2005*)). Under the following restrictions:

$$I_{up}(\alpha) \equiv \{t|\alpha_t < C, y_t = 1 \text{ or } \alpha_t > 0, y_t = -1\}, \tag{2}$$
$$I_{low}(\alpha) \equiv \{t|\alpha_t < C, y_t = -1 \text{ or } \alpha_t > 0, y_t = 1\}. \tag{3}$$

For the $k^{th}$ iteration, if $i \in I_{up}(\alpha^k), j \in I_{low}(\alpha^k)$, and $-y_i \nabla f(\alpha^k)_i > -y_j \nabla f(\alpha^k)_j$, then $\{i, j\}$ is a "violating pair".

Violating pairs are important in WSS. If working set $B$ is a violating pair, the function value in SMO-type decomposition methods strictly decreases (*Hush & Scovel, 2003*). Under the definition of violating pair, a natural choice of the working set $B$ is the "maximal violating pair", which most violates the KKT optimality condition.

**WSS 1** (WSS *via* the "maximal violating pair" (*Keerthi, Shevade & Bhattacharyya, 2001*; *Fan, Chen & Lin, 2005*; *Chen, Fan & Lin, 2006*)). Under the same restrictions (2) and (3) in Definition 1,

1. Select

$$i \in \arg\max_t \{-y_t \nabla f(\alpha^k)_t | t \in I_{up}(\alpha^k)\}, \tag{4}$$

$$j \in \arg\min_t \{-y_t \nabla f(\alpha^k)_t | t \in I_{low}(\alpha^k)\}, \tag{5}$$

or

$$j \in \arg\max_t \{y_t \nabla f(\alpha^k)_t | t \in I_{low}(\alpha^k)\}. \tag{6}$$

2. Return $B = \{i, j\}$.

*Keerthi, Shevade & Bhattacharyya (2001)* first proposed the maximal violating pair, which has become a popular way in WSS. *Fan, Chen & Lin (2005)* pointed out that it was concerned with the first order approximation of $f(\alpha)$ in (1) and gave a detailed explanation. Meanwhile, they proposed a new WSS algorithm by using more accurate second order information.

**WSS 2** (WSS using second order information (*Fan, Chen & Lin, 2005*; *Chen, Fan & Lin, 2006*)).

1. Define $a_{it}$ and $b_{it}$,

$$a_{it} \equiv K_{ii} + K_{tt} - 2K_{it}, \tag{7}$$

$$b_{it} \equiv -y_i\nabla f(\alpha^k)_i + y_t\nabla f(\alpha^k)_t > 0, \tag{8}$$

$$\bar{a}_{it} \equiv \begin{cases} a_{it} & \text{if } a_{it} > 0, \\ \tau & \text{otherwise.} \end{cases} \tag{9}$$

2. Select

$$i \in \arg\max_t\{-y_t\nabla f(\alpha^k)_t | t \in I_{up}(\alpha^k)\}, \tag{10}$$

$$j \in \arg\min_t\left\{-\frac{b_{it}^2}{\bar{a}_{it}} | t \in I_{low}(\alpha^k), -y_t\nabla f(\alpha^k)_t < -y_i\nabla f(\alpha^k)_i\right\}. \tag{11}$$

3. Return $B = \{i, j\}$.

WSS 2 uses second order information and checks only $O(l)$ possible working sets to select $j$ through using the same $i$ as in WSS 1. The WSS 2 algorithm achieves faster convergence than existing selection methods using first order information. It has been used in the software LIBSVM (*Chang & Lin, 2007*) (since version 2.8) and is valid for all symmetric kernel matrices $K$, including the non-positive definite kernel.

*Lin (2001, 2002)* pointed out the maximal violating pair was important to SMO-type methods. When the working set $B$ is the maximal violating pair, SMO-type methods converge to a stationary point. Otherwise, it is uncertain whether the convergence will be established. *Chen, Fan & Lin (2006)* proposed a general WSS method *via* the "constant-factor violating pair". Under a fixed constant-factor $\sigma$ specified by the user, the selected violating pair is linked to the maximal violating pair. The "constant-factor violating pair" is considered to be a "sufficiently violated" pair. And they prove the convergence of the WSS method.

**WSS 3** (WSS *via* the "constant-factor violating pair" (*Fan, Chen & Lin, 2005*; *Chen, Fan & Lin, 2006*)).
1. Given a fixed $0 < \sigma \leq 1$ in all iterations.
2. Compute

$$m(\alpha^k) = \max_t\{-y_t\nabla f(\alpha^k)_t | t \in I_{up}(\alpha^k)\}, \tag{12}$$

$$M(\alpha^k) = \min_t\{-y_t\nabla f(\alpha^k)_t | t \in I_{low}(\alpha^k)\}. \tag{13}$$

3. Select $i, j$ satisfying

$$i \in I_{up}(\alpha^k), \quad j \in I_{low}(\alpha^k), \tag{14}$$

$$-y_i\nabla f(\alpha^k)_i + y_j\nabla f(\alpha^k)_j \geq \sigma(m(\alpha^k) - M(\alpha^k)) > 0. \tag{15}$$

4. Return $B = \{i, j\}$.

Clearly (15) guarantees the quality of the working set $B$ if it is related to the maximal violating pair. *Fan, Chen & Lin (2005)* explained that WSS 2 was a special case of WSS 3 under the special value of $\sigma$.

Furthermore, *Zhao et al. (2007)* employed algorithm WSS 2 to test the datasets by LIBSVM. They find two interesting phenomena. One is that some $\alpha$ are not updated in the entire training process. Another is that some $\alpha$ are updated again and again. Therefore, they propose a new method WSS-WR and a certain $\alpha$ are selected only once to improve the efficiency of WSS, especially the reduction of the training time.

## Differential privacy

Recently, with the advent of the digital age, huge amounts of personal information have been collected by web services and mobile devices. Although data sharing and mining large-scale personal information can help improve the functionality of these services, it also raises privacy concerns for data contributors. DP provides a mathematically rigorous definition of privacy and has become a new accepted standard for private data analysis. It ensures that any possible outcome of an analysis is almost equal regardless of an individual's presence or absence in the dataset, and the output difference is controlled by a relatively small privacy budget. The smaller the budget, the higher the privacy. Therefore, the adversary cannot distinguish whether an individual's in the dataset (*Liu, Li & Li, 2017*). Furthermore, DP is compatible with various kinds of data sources, data mining algorithms, and data release models.

In dataset $D$, each row corresponds to one individual, and each column represents an attribute value. If two datasets $D$ and $D'$ only differ on one element, they are defined as neighboring datasets. DP aims to mask the different results of the query function $f$ in neighboring datasets. The maximal difference of the query results is defined as the sensitivity $\Delta f$. DP is generally achieved by a randomized mechanism $M : D \rightarrow R^d$, which returns a random vector from a probability distribution. A mechanism $M$ satisfies DP if the effect of the outcome probability by adding or removing a single element is controlled within a small multiplicative factor (*Lee, 2014*). The formal definition is given as follows.

**Definition 2** ($\varepsilon$-differential privacy (*Dwork, 2006*)). A randomized mechanism $M$ gives $\varepsilon$-DP if for all datasets $D$ and $D'$ differing on at most one element, and for all subsets of possible outcomes $S \subseteq$ Range $(M)$,

$$\Pr[M(D) \in S] \leq \exp(\varepsilon) \times \Pr[M(D') \in S]. \tag{16}$$

Sensitivity is a vital concept in DP that represents the largest effect of the query function output made by a single element. Meanwhile, sensitivity determines the requirements of how much perturbation by a particular query function (*Zhu et al., 2017*).

**Definition 3** (Sensitivity (*Dwork, 2006*)). For a given query function $f : D \rightarrow R^d$, and neighboring datasets $D$ and $D'$, the sensitivity of $f$ is defined as

$$\Delta f = \max_{D,D'} \|f(D) - f(D')\|_1. \tag{17}$$

The sensitivity $\Delta f$ depends only on the query function $f$, and not on the instances in datasets.

Any mechanism that meets Definition 2 is considered as satisfying DP (*Lee, 2014*). Currently, two principal mechanisms have been used for realizing DP: the Laplace mechanism (*Dwork et al., 2006*) and the exponential mechanism (*McSherry & Talwar, 2007*).

**Definition 4** (Laplace mechanism (*Dwork et al., 2006*)). For a numeric function $f : D \rightarrow R^d$ on a dataset $D$, the mechanism $M$ in Eq. (18) provides $\varepsilon$-DP.

$$M(D) = f(D) + Lap\left(\frac{\Delta f}{\varepsilon}\right)^d. \tag{18}$$

The Laplace mechanism gets the real results from the numerical query and then perturbs it by adding independent random noise. Let $Lap(b)$ represent the random noise sampled from a Laplace distribution according to sensitivity. The Laplace mechanism is usually used for numerical data, while for the non-numerical queries, DP uses the exponential mechanism to randomize results.

**Definition 5** (Exponential mechanism (*McSherry & Talwar, 2007*)). Let $q(D, r)$ be a scoring function on a dataset $D$ that measures the quality of output $r \in R$, $\Delta q$ represents the sensitivity. The mechanism $M$ satisfies $\varepsilon$-DP if

$$M(D) = \left(return \ r \propto \exp\left(\frac{\varepsilon q(D,r)}{2\Delta q}\right)\right). \tag{19}$$

The exponential mechanism is useful to select a discrete output in a differentially private manner, which employs a scoring function $q$ to evaluate the quality of an output $r$ with a nonzero probability.

# DPWSS ALGORITHM

In this paper, we study the problem of how to privately release the classification model of SVMs while satisfying DP. To overcome the shortcomings of the privacy-preserving SVM classification methods, such as low accuracy or complex sensitivity analysis of output perturbation and objective perturbation, we proposed the algorithm DPWSS for training SVM in this section. The DPWSS algorithm is achieved by privately selecting the working set with the exponential mechanism in every iteration. As far as we know, DPWSS is the first private WSS algorithm based on DP.

## An improved WSS method

In the process of training SVMs, WSS is an important step in SMO-type decomposition methods. Meanwhile, the special properties of the selection process in WSS are perfectly combined with the exponential mechanism of DP. WSS 3 algorithm is a more general algorithm to select a working set by checking nearly $O(l^2)$ possible $B$'s to decide $j$, although

under the restricted condition of parameter $\sigma$. By using the same $i \in \arg m(a^k)$ as in WSS 2, which checks only $O(l)$ possible $B$'s, we propose WSS 4 to select a working set based on WSS 3 as below. To make the algorithm easy to understand, we replace $M(a^k)$ with $M'(a^k)$.

**WSS 4** (An improved WSS *via* the "constant-factor violating pair")
1. Given a fixed $0 < \sigma \leq 1$ in all iterations.
2. Compute

$$m(\alpha^k) = \max_t \{-y_t \nabla f(\alpha^k)_t | t \in I_{up}(\alpha^k)\}, \tag{20}$$

$$M'(\alpha^k) = \max_t \{y_t \nabla f(\alpha^k)_t | t \in I_{low}(\alpha^k)\}. \tag{21}$$

3. Select $i, j$ satisfying

$$i \in \arg m(\alpha^k), \quad j \in I_{low}(\alpha^k), \tag{22}$$
$$m(\alpha^k) + y_j \nabla f(\alpha^k)_j \geq \sigma(m(\alpha^k) + M'(\alpha^k)) > 0. \tag{23}$$

4. Return $B = \{i, j\}$.

## The scoring function and sensitivity in the exponential mechanism

In the exponential mechanism, the scoring function is an important guarantee for achieving DP. The rationality of scoring function design is directly related to the execution efficiency of mechanism $M$. For one output $r$, the greater the value of the scoring function, the greater the probability that $r$ will be selected. Based on the definition of the "maximal violating pair", it is obvious that

$$m(\alpha^k) + M'(\alpha^k) \geq m(\alpha^k) + y_j \nabla f(\alpha^k)_j. \tag{24}$$

From Eqs. (23) and (24), we conclude that

$$m(\alpha^k) + M'(\alpha^k) \geq m(\alpha^k) + y_j \nabla f(\alpha^k)_j \geq \sigma(m(\alpha^k) + M'(\alpha^k)) > 0. \tag{25}$$

We design a simple scoring function $q(D, r)$ for the DPWSS algorithm based on WSS 4 and Eq. (25) as follows

$$1 \geq q(D, r) = \frac{m(\alpha^k) + y_j \nabla f(\alpha^k)_j}{m(\alpha^k) + M'(\alpha^k)} \geq \sigma, \tag{26}$$

where $r$ denotes the working set $B$, which contains violating pair $i$ and $j$. The larger the value of scoring function $q(D, r)$, the closer the selected violation pair is to the maximal violation pair. The sensitivity of scoring function $q(D, r)$ is

$$\Delta q = 1 - \sigma, \tag{27}$$

and the value of $\Delta q$ is a small number, less than 1.

In the exponential mechanism, the output $r$ is selected randomly with probability

$$\Pr(r) = \exp\left(\frac{\varepsilon q(D, r)}{2\Delta q}\right) \Bigg/ \sum_{r' \in R} \exp\left(\frac{\varepsilon q(D, r')}{2\Delta q}\right). \tag{28}$$

## Privacy budget

Privacy budget is a vital parameter in DP, which controls the privacy level in a randomized mechanism $M$. The smaller the privacy budget, the higher the privacy level. When the allocated privacy budget runs out, mechanism $M$ will lose privacy protection, especially for the iteration process. To improve the utilization of the privacy budget, every pair of working sets is selected only once during the entire training process as in *Zhao et al. (2007)*. Meanwhile, in DPWSS every iteration is based on the result of the last iteration, but not based on the entire original dataset. Therefore, there is no need to split the privacy budget for every iteration.

## Description of DPWSS algorithm

In the DPWSS algorithm, DP is achieved by privately selecting the working set with the exponential mechanism in every iteration. We first present an overview of the DPWSS algorithm and then elaborate on the key steps. Finally, we describe an SMO-type decomposition method using the DPWSS algorithm in detail.

The description of the DPWSS Algorithm 1 is shown below.

The DPWSS algorithm selects multiple violating pairs that meet the constraints based on WSS 4, and then randomly selects one with a certain probability by the exponential mechanism to satisfy DP. Firstly, the DPWSS algorithm computes $m(\alpha)$ and $M'(\alpha)$ for the scoring function $q$ from Line 1 to Line 4 and determines $i$ as one element of the violating pair. Secondly, it computes the scoring function $q$ from Line 5 to Line 12. The constraints in Line 6 represent that the violating pair $\{i, j\}$ has not been previously selected, meanwhile the value range of the other element $j$ and the violating pair are valid for the changes of gradient $G$. The constraints in Line 8 represent that the scoring function value is effective under constant-factor $\sigma$. Line 14 and Line 15 are key steps in the exponential mechanism, which randomly select a violating pair with the chosen probability of the scoring function $q$. Lastly, the DPWSS algorithm outputs the violating pair $\{i, j\}$ as the working set $B$ in Line 15. The time and memory complexity of DPWSS algorithm is $O(l)$.

In summary, a SMO method using the DPWSS algorithm is shown below.

Algorithm 2 is an iterative process, which first selects working set $B$ by DPWSS, then updates dual vector $\alpha$ and gradient $G$ in every iteration. After the iterative process, the algorithm outputs the final $\alpha$. There are three ways to get out of the iterative process. One is that $\alpha$ is a stationary point, another is that all violating pairs have been selected, and the last one is that the number of iterations exceeds the maximum value. Using Algorithm 2, we privately release the classification model of SVMs with dual vector $\alpha$ while satisfying the requirement of DP.

---

**Algorithm 1 DPWSS.**

**Input:** $G$: gradient array; $y$: array of every instance labels with $\{+1, -1\}$; $l$: number of instances; $\alpha$: dual vector; $I$: the violating pair selected flag bool matrix; $\sigma$: constant-factor; $\varepsilon$: privacy budget; $eps$: stopping tolerance;

**Output:** $B$: working set;

**Begin**

1:   initialize $m(\alpha)$ and $M'(\alpha)$   to -INF;

2:   find $m(\alpha)$ by Eq. (20) for $t$ in $[0{:}l{-}1]$ and $t$ in $I_{up}(\alpha)$;

3:   set $i = t$;

4:   find $M'(\alpha)$ by Eq. (21) for $t$ in $[0{:}l{-}1]$ and $t$ in $I_{low}(\alpha)$;

5:   for $t = 0$ to $l{-}1$

6:          if $I[i][t] ==$false and $t$ in $I_{low}(\alpha)$ and $m(\alpha)+y[t]^*G[t] > eps$ then

7:                 compute scoring function $q(D, r_t)$ by Eq. (26);

8:                 if $q(D, r_t) \geq \sigma$ then

9:                        $q'(D, r_t) \leftarrow q(D, r_t)$;

10:                end if

11:       end if

12:   end for

13:   compute the probability $\Pr(B)$ for every violating pair by Eq. (28);

14:   randomly select a violating pair as a working set with probability $\Pr(B)$;

15:   Return $B = \{i, j\}$;

**End**

---

## Privacy analysis

In the DPWSS algorithm, randomness is introduced by randomly selecting working sets with the exponential mechanism. By using the exponential mechanism, a violating pair is selected randomly with a certain probability. The greater the probability, the closer the selected violating pair is to the maximal violating pair. For every iteration, the violating pair in the outputs of the DPWSS algorithm is uncertain. The uncertainty masks the impact of individual record change on the algorithm results, thus protecting the data privacy.

According to the definition of DP mentioned in Section 3, we proved that the DPWSS algorithm satisfies DP strictly by theorem 1 as shown below.

**Theorem 1** DPWSS algorithm satisfies DP.

**Proof** Let $M(D, q)$ be to select the output $r$ of the violating pair in one iteration, and $\varepsilon$ be the allocated privacy budget in the DPWSS algorithm. Based on Eq. (28), we randomly select violating pair $r$ as a working set with the following probability by the exponential mechanism. To accord with the standard form of the exponential mechanism, we use $q$ to denote $q'$ in the DPWSS algorithm.

---

**Algorithm 2** A SMO method using DPWSS.

---

**Input:** $Q$: kernel symmetric matrix; $y$: array of every instance labels with $\{+1, -1\}$; $l$: number of instances; $C$: upper bound of all dual variables;

**Output:** $\alpha$: dual vector;

**Begin**

1:    initialize gradient array $G$ to all $-1$, dual vector $\alpha$ to all 0, and violating pair selected flagbool matrix  $I$ to all 0;

2:    find $\alpha^I$ as the initial feasible solution, set $k = 1$;

3:    **while** $k <$ max_iter

4:    **if** $\alpha^k$ is a stationary point then

5:        exit the loop;

6:    **else**

7:        select working set $B = \{i, j\}$ by DPWSS;

8:        if $B$ is *NULL* then

9:          exit the loop;

10:       end if

11:    **end if**

12:    set $k = k+1$;

13:    set $I[i][j] =$ true;

14:    update $\alpha[i]$ and $\alpha[j]$;

15:    project $\alpha$ back to the feasible region;

16:    update gradient $G$;

17:    **end while**

18:    return $\alpha$;

**End**

---

$$\frac{\Pr(M(D, q) = r)}{\Pr(M(D', q) = r)} = \frac{\exp\left(\frac{\varepsilon q(D, r)}{2\Delta q}\right) / \sum_{r' \in O} \exp\left(\frac{\varepsilon q(D, r')}{2\Delta q}\right)}{\exp\left(\frac{\varepsilon q(D', r)}{2\Delta q}\right) / \sum_{r' \in O} \exp\left(\frac{\varepsilon q(D', r')}{2\Delta q}\right)}$$

$$= \exp\left(\frac{\varepsilon(q(D, r) - q(D', r))}{2\Delta q}\right) \times \frac{\sum_{r' \in O} \exp\left(\frac{\varepsilon q(D', r')}{2\Delta q}\right)}{\sum_{r' \in O} \exp\left(\frac{\varepsilon q(D, r')}{2\Delta q}\right)}$$

$$\leq \exp\left(\frac{\varepsilon}{2}\right) \times \frac{\sum_{r' \in O} \exp\left(\frac{\varepsilon}{2}\right) \exp\left(\frac{\varepsilon q(D, r')}{2\Delta q}\right)}{\sum_{r' \in O} \exp\left(\frac{\varepsilon q(D, r')}{2\Delta q}\right)}$$

**Table 2 Basic information of the datasets.**

| Index | Dataset | #data | Range | #features | Imbalance ratio |
|---|---|---|---|---|---|
| 1 | a1a | 1,605 | (0,1) | 119 | 0.33 |
| 2 | a5a | 6,414 | (0,1) | 122 | 0.32 |
| 3 | Australian | 690 | (−1,1) | 14 | 0.8 |
| 4 | breast | 683 | (−1,1) | 10 | 1.86 |
| 5 | diabetes | 768 | (−1,1) | 8 | 1.87 |
| 6 | fourclass | 862 | (−1,1) | 2 | 0.55 |
| 7 | German | 1,000 | (−1,1) | 24 | 0.43 |
| 8 | gisette | 6,000 | (−1,1) | 5,000 | 1 |
| 9 | heart | 270 | (−1,1) | 13 | 0.8 |
| 10 | ijcnn1 | 49,990 | (−1,1) | 22 | 0.11 |
| 11 | ionosphere | 351 | (−1,1) | 34 | 1.79 |
| 12 | rcv1 | 20,242 | (−1,1) | 47,236 | 1.08 |
| 13 | sonar | 208 | (−1,1) | 60 | 0.87 |
| 14 | splice | 1,000 | (−1,1) | 60 | 1.07 |
| 15 | w1a | 2,477 | (0,1) | 300 | 0.03 |
| 16 | w5a | 9,888 | (0,1) | 300 | 0.03 |

$$= \exp\left(\frac{\varepsilon}{2}\right) \times \exp\left(\frac{\varepsilon}{2}\right) \times \frac{\sum_{r' \in O} \exp\left(\frac{\varepsilon q(D, r')}{2\Delta q}\right)}{\sum_{r' \in O} \exp\left(\frac{\varepsilon q(D, r')}{2\Delta q}\right)} = \exp(\varepsilon)$$

According to Definition 2, we prove that

$$\Pr(M(D, q) = r) \leq \exp(\varepsilon) \times \Pr(M(D', q) = r).$$

Therefore, the DPWSS algorithm satisfies DP.

Algorithm 2 is an iterative process, in which DPWSS is a vital step to privately select a working set. As the DPWSS algorithm satisfies DP, we perform the steps of updating dual vector $\alpha$ and gradient $G$ in every iteration without accessing private data. To improve the utilization of the privacy budget, every pair of working sets is selected only once during the entire training process. Meanwhile, in Algorithm 2 every iteration is based on the result of the last iteration, but not based on the original datasets. Therefore, Algorithm 2 satisfies DP.

## EXPERIMENTS

In this section, we compared the performance of the DPWSS algorithm with WSS 2, which is a classical non-private WSS algorithm and has been used in the software LIBSVM (*Chang & Lin, 2007*). The comparison between WSS 2 and WSS 1 was done in *Fan, Chen & Lin (2005)*. We do not compare the DPWSS algorithm with other private SVMs. One reason is that randomness is introduced in different ways, and the other reason is that

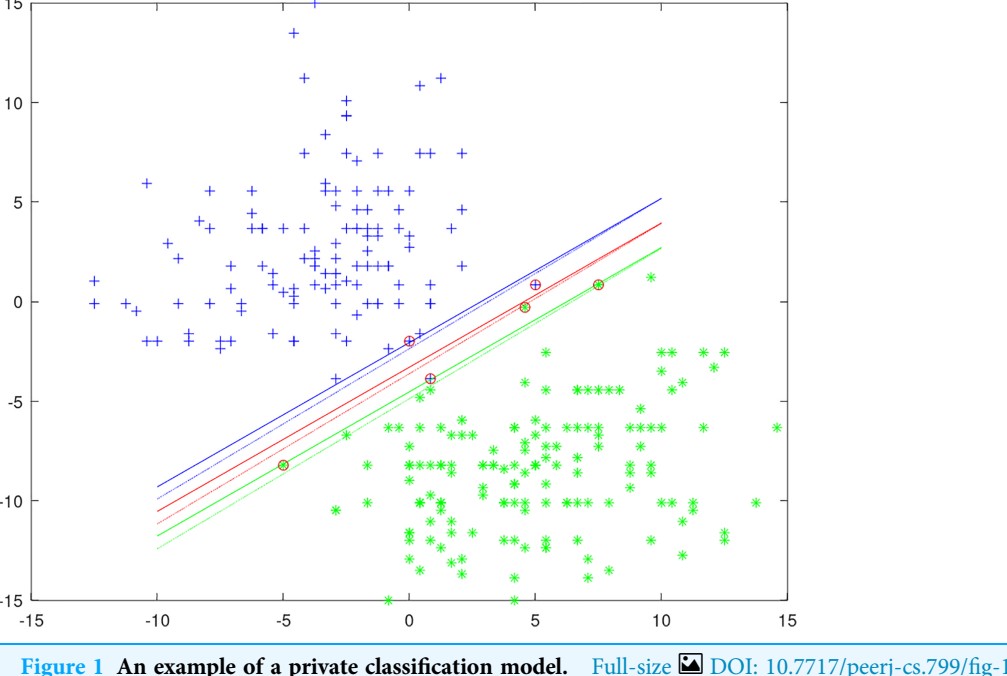

**Figure 1 An example of a private classification model.**

the DPWSS algorithm achieves classification accuracy and optimized objective value almost the same as the original non-privacy SVM algorithm.

## Datasets and experimental environment

The datasets are partly selected for the experiments as *Zhang, Hao & Wang (2019)*, *Fan, Chen & Lin (2005)* and *Zhao et al. (2007)*. All datasets are for binary classification, and available at http://www.csie.ntu.edu.tw/~cjlin/libsvmtools/. The basic information of the datasets includes dataset size, value range, number of features, and imbalance ratio, which is shown in Table 2 below. To make the figures look neater in the experiments, we use breast to denote the breast-cancer dataset and German to denote the german.number dataset.

To carry out the contrast experiments efficiently, we use LIBSVM (version 3.24) as an implementation of the DPWSS algorithm in C++ language and GNU Octave (version 5.2). All parameters are set to default values.

## An example of a private classification model

Unlike other privacy SVMs, which introduce randomness into the objective function or classification result by the Laplace mechanism, in our method the randomness is introduced into the training process of SVM. It is achieved by privately selecting the working set with the exponential mechanism in every iteration. We give an example of a private classification model to show how privacy is protected in Fig. 1. The data uses two columns of the heart dataset and moves the positive and negative instances to each end for easier classification. The solid lines represent the original non-private classification model and circles represent support vectors. The dotted lines represent a private

classification model by training SVM with the DPWSS algorithm. It is observed that the differences between the private and non-private classification models are very small, and achieves similar accuracy of classification. All the classification models generated are different from each other to protect the training data privacy.

## Algorithm performance experiments

In this section, we evaluated the performance of the DPWSS algorithm *vs* WSS 2 by experiments for the entire training process. The metrics of performance include classification capability, algorithm stability, and execution efficiency under different constant-factor $\sigma$ and privacy budget $\varepsilon$.

The classification capability is measured by *AUC*, *Accuracy*, *Precision*, *Recall*, *F1*, and *Mcc*.

$$AUC = \frac{\sum_{i \in positiveClass} rank_i - \frac{M(1+M)}{2}}{M \times N} \tag{29}$$

The $rank_i$ denotes the serial number of instance $i$ after sorting by the probability, $M$ is the number of positive instances and $N$ is the number of negative instances. The higher the *AUC*, the better the usability of the algorithm. Other metrics are calculated as shown below, and they are all based on confusion matrix.

$$Accuracy = \frac{TP + TN}{TP + TN + FP + FN} \tag{30}$$

$$Precision = \frac{TP}{TP + FP} \tag{31}$$

$$Recall = \frac{TP}{TP + FN} \tag{32}$$

$$F1 = \frac{2TP}{2TP + FP + FN} \tag{33}$$

$$Mcc = \frac{TP \times TN - FP \times FN}{\sqrt{(TP + FP)(TP + FN)(TN + FP)(TN + FN)}} \tag{34}$$

The algorithm stability is measured by the error of optimized objective value between DPWSS algorithm and WSS 2, named *objError*.

$$objError = |obj_{DPWSS} - obj_{WSS2}| \tag{35}$$

The smaller the *objError*, the better the stability of the algorithm.

The execution efficiency of the algorithm is measured by the ratio of iteration between the two algorithms, named *iterationRatio*.

$$iterationRatio = \frac{\#iteration \text{ with } DPWSS}{\#iteration \text{ with } WSS2} \tag{36}$$

The smaller the *iterationRatio*, the better the execution efficiency of the algorithm. We do not compare the training time between the two algorithms as it is a millisecond class for the entire training process to most of the datasets.

To evaluate the influence of different constant-factor $\sigma$ and privacy budget $\varepsilon$ to the three metrics for algorithm performance, we set $\sigma$ at 0.1, 0.3, 0.5 and 0.7 under $\varepsilon$ fixed at 1 and set $\varepsilon$ at 0.1, 0.5 and 1 under $\sigma$ fixed at 0.7. We do not set $\sigma$ at 0.9, because under the circumstances, most of the violating pairs will be filtered out that the algorithm fails to reach the final objective value.

Firstly, we measure the classification capability of the DPWSS algorithm *vs* WSS 2. The experiments for the DPWSS algorithm were repeated five times under different $\sigma$ and $\varepsilon$, and the averages of the experimental results are shown in Table 3. Observed from the results, the DPWSS algorithm achieves almost the same classification capability as WSS 2 on all datasets. The maximum error between them is no more than 3%. Due to the repeated execution of the iterative process, the DPWSS algorithm obtains a well private classification model. The classification capability is not affected by the randomness of DP and the filtering effect of parameter $\sigma$ on violating pairs. The DPWSS algorithm introduces randomness into the training process of SVMs, not into the objective function or classification result. There are no requirements of the differentiability of the objective function and the complex sensitivity analysis, and the less influence of high-dimensional data on noise. Therefore, the DPWSS algorithm achieves the target extremum through the optimization process under the current condition. Meanwhile, the imbalance of dataset has little effect on the classification capability of the DPWSS algorithm.

Secondly, we compare the optimized objective values and measure the algorithm stability by *objError* between the DPWSS algorithm and WSS 2. The experimental results are shown in Figs. 2–5. Observed from the results, the DPWSS algorithm achieves similar optimized objective values with WSS 2 on all datasets under different $\sigma$ and $\varepsilon$. The errors between the DPWSS algorithm and WSS 2 are very small (nearly within two). Due to the repeated execution of the iterative process, the DPWSS algorithm converges stably to optimized objective values and is not affected by the randomness of DP and the filtering effect of parameter $\sigma$ on violating pairs. With the increase of $\sigma$, the errors also tends to increase.

Lastly, we compare the iterations and measure the execution efficiency by *iterationRatio* between the two algorithms. The experimental results are shown in Figs. 6–21. Observed from the results, the DPWSS algorithm achieves higher execution efficiency with fewer iterations *vs* WSS 2 on all datasets under different $\sigma$ and $\varepsilon$. Because the DPWSS algorithm introduces randomness into the WSS process, the iterations will increase more or less. However, with the increase of constant-factor $\sigma$, the iterations are affected by the filtering effect of it on violating pairs larger and larger. When $\sigma$ increases to 0.3, the execution efficiency of the DPWSS algorithm is already higher than WSS 2 for most datasets. When $\sigma$ increases to 0.7, the iterations of the DPWSS algorithm are far less than WSS2 for all datasets except ijcnn1. Therefore, our method should set larger $\sigma$ for big datasets. While the privacy budget $\varepsilon$ has little effect on iterations under a fixed constant-factor $\sigma$.

**Table 3 The performance of WSS2 and DPWSS for different ε and σ.**

| Dataset | Shrinking | Metrics | WSS2 | DPWSS | | | | | |
|---|---|---|---|---|---|---|---|---|---|
| | | | | epsi = 1 sigm = 0.1 | epsi = 1 sigm = 0.3 | epsi = 1 sigm = 0.5 | epsi = 1 sigm = 0.7 | epsi = 0.5 sigm = 0.7 | epsi = 0.1 sigm = 0.7 |
| a1a | 1 | AUC | 0.9117 | 0.9119 | 0.9117 | 0.9113 | 0.9116 | 0.9109 | 0.9123 |
| | | Accuracy | 0.8623 | 0.8611 | 0.8629 | 0.8629 | 0.8592 | 0.8604 | 0.8636 |
| | | Precision | 0.7669 | 0.7654 | 0.7709 | 0.7709 | 0.76 | 0.7631 | 0.7667 |
| | | Recall | 0.6329 | 0.6278 | 0.6304 | 0.6304 | 0.6253 | 0.6278 | 0.6405 |
| | | F1 | 0.6935 | 0.6898 | 0.6936 | 0.6936 | 0.6861 | 0.6889 | 0.6979 |
| | | Mcc | 0.6104 | 0.6063 | 0.6115 | 0.6115 | 0.6012 | 0.6048 | 0.6148 |
| | | obj | −540.57 | −540.42 | −540.45 | −540.42 | −539.95 | −540.23 | −540.06 |
| | | iteration | 8,649 | 10,535 | 6,735 | 4,316 | 3,239 | 3,529 | 3,406 |
| | 0 | AUC | 0.9117 | 0.9116 | 0.9116 | 0.9119 | 0.9122 | 0.9111 | 0.9117 |
| | | Accuracy | 0.8623 | 0.8623 | 0.8629 | 0.8629 | 0.8617 | 0.8623 | 0.8617 |
| | | Precision | 0.7669 | 0.7685 | 0.7692 | 0.7709 | 0.7645 | 0.7685 | 0.7645 |
| | | Recall | 0.6329 | 0.6304 | 0.6329 | 0.6304 | 0.6329 | 0.6304 | 0.6329 |
| | | F1 | 0.6935 | 0.6926 | 0.6944 | 0.6936 | 0.6925 | 0.6926 | 0.6925 |
| | | Mcc | 0.6104 | 0.61 | 0.612 | 0.6115 | 0.6088 | 0.61 | 0.6088 |
| | | obj | −540.57 | −540.42 | −540.44 | −540.38 | −540.12 | −540.11 | −540.2 |
| | | iteration | 7,997 | 9,566 | 6,091 | 4,252 | 3,447 | 3,379 | 3,295 |
| a5a | 1 | AUC | 0.906 | 0.9059 | 0.9059 | 0.9057 | 0.9058 | 0.9058 | 0.9059 |
| | | Accuracy | 0.8506 | 0.8502 | 0.8499 | 0.8511 | 0.8486 | 0.8506 | 0.8503 |
| | | Precision | 0.7327 | 0.7317 | 0.7306 | 0.7354 | 0.7311 | 0.7334 | 0.7305 |
| | | Recall | 0.6131 | 0.6119 | 0.6119 | 0.6112 | 0.6029 | 0.6119 | 0.615 |
| | | F1 | 0.6676 | 0.6664 | 0.666 | 0.6676 | 0.6608 | 0.6671 | 0.6678 |
| | | Mcc | 0.576 | 0.5746 | 0.5738 | 0.5768 | 0.5689 | 0.5758 | 0.5757 |
| | | obj | −2,224.72 | −2,224.56 | −2,224.57 | −2,224.33 | −2,223.2 | −2,223.75 | −2,223.92 |
| | | iteration | 35,752 | 36,151 | 22,590 | 15,987 | 13,240 | 14,034 | 14,275 |
| | 0 | AUC | 0.906 | 0.9058 | 0.9059 | 0.9058 | 0.9057 | 0.9058 | 0.906 |
| | | Accuracy | 0.8506 | 0.8506 | 0.8506 | 0.8514 | 0.8503 | 0.8502 | 0.85 |
| | | Precision | 0.7327 | 0.7323 | 0.7327 | 0.7373 | 0.7312 | 0.7331 | 0.7322 |
| | | Recall | 0.6131 | 0.6138 | 0.6131 | 0.6099 | 0.6138 | 0.6093 | 0.6099 |
| | | F1 | 0.6676 | 0.6678 | 0.6676 | 0.6676 | 0.6674 | 0.6655 | 0.6655 |
| | | Mcc | 0.576 | 0.5762 | 0.576 | 0.5773 | 0.5754 | 0.5741 | 0.5738 |
| | | obj | −2,224.72 | −2,224.41 | −2,224.47 | −2,224.33 | −2,223.72 | −2,223.07 | −2,223.85 |
| | | iteration | 37,578 | 33,682 | 21,592 | 16,418 | 13,926 | 12,987 | 14,318 |
| Australian | 1 | AUC | 0.9393 | 0.9403 | 0.9378 | 0.9318 | 0.9141 | 0.9126 | 0.9202 |
| | | Accuracy | 0.8565 | 0.8565 | 0.8565 | 0.8565 | 0.8565 | 0.8565 | 0.8565 |
| | | Precision | 0.7873 | 0.7873 | 0.7873 | 0.7873 | 0.7873 | 0.7873 | 0.7873 |
| | | Recall | 0.9283 | 0.9283 | 0.9283 | 0.9283 | 0.9283 | 0.9283 | 0.9283 |
| | | F1 | 0.852 | 0.852 | 0.852 | 0.852 | 0.852 | 0.852 | 0.852 |
| | | Mcc | 0.7237 | 0.7237 | 0.7237 | 0.7237 | 0.7237 | 0.7237 | 0.7237 |
| | | obj | −199.65 | −199.25 | −198.98 | −198.21 | −197.78 | −198.53 | −198.64 |
| | | iteration | 10,727 | 6,438 | 1,910 | 835 | 493 | 596 | 612 |

*(Continued)*
| Table 3 (continued) | | | | | | | | | |
|---|---|---|---|---|---|---|---|---|---|
| Dataset | Shrinking | Metrics | WSS2 | DPWSS | | | | | |
| | | | | epsi = 1 sigm = 0.1 | epsi = 1 sigm = 0.3 | epsi = 1 sigm = 0.5 | epsi = 1 sigm = 0.7 | epsi = 0.5 sigm = 0.7 | epsi = 0.1 sigm = 0.7 |
| | 0 | AUC | 0.9393 | 0.9397 | 0.9324 | 0.9111 | 0.923 | 0.9223 | 0.9316 |
| | | Accuracy | 0.8565 | 0.8565 | 0.8565 | 0.8565 | 0.8565 | 0.8565 | 0.8565 |
| | | Precision | 0.7873 | 0.7873 | 0.7873 | 0.7873 | 0.7873 | 0.7873 | 0.7873 |
| | | Recall | 0.9283 | 0.9283 | 0.9283 | 0.9283 | 0.9283 | 0.9283 | 0.9283 |
| | | F1 | 0.852 | 0.852 | 0.852 | 0.852 | 0.852 | 0.852 | 0.852 |
| | | Mcc | 0.7237 | 0.7237 | 0.7237 | 0.7237 | 0.7237 | 0.7237 | 0.7237 |
| | | obj | −199.65 | −199.25 | −199.15 | −198.68 | −198.33 | −198.62 | −198.82 |
| | | iteration | 10,590 | 6,978 | 2,629 | 847 | 542 | 637 | 731 |
| breast | 1 | AUC | 0.9962 | 0.9962 | 0.9963 | 0.9961 | 0.9962 | 0.9961 | 0.9962 |
| | | Accuracy | 0.9707 | 0.9707 | 0.9707 | 0.9707 | 0.9707 | 0.9707 | 0.9707 |
| | | Precision | 0.9818 | 0.9818 | 0.9818 | 0.9818 | 0.9818 | 0.9818 | 0.9818 |
| | | Recall | 0.973 | 0.973 | 0.973 | 0.973 | 0.973 | 0.973 | 0.973 |
| | | F1 | 0.9774 | 0.9774 | 0.9774 | 0.9774 | 0.9774 | 0.9774 | 0.9774 |
| | | Mcc | 0.936 | 0.936 | 0.936 | 0.936 | 0.936 | 0.936 | 0.936 |
| | | obj | −46 | −45.96 | −45.93 | −45.89 | −45.63 | −45.53 | −45.78 |
| | | iteration | 212 | 542 | 257 | 196 | 138 | 146 | 150 |
| | 0 | AUC | 0.9962 | 0.9962 | 0.9963 | 0.9963 | 0.9962 | 0.9962 | 0.9962 |
| | | Accuracy | 0.9707 | 0.9707 | 0.9707 | 0.9707 | 0.9722 | 0.9722 | 0.9722 |
| | | Precision | 0.9818 | 0.9818 | 0.9818 | 0.9818 | 0.9819 | 0.9841 | 0.9819 |
| | | Recall | 0.973 | 0.973 | 0.973 | 0.973 | 0.9752 | 0.973 | 0.9752 |
| | | F1 | 0.9774 | 0.9774 | 0.9774 | 0.9774 | 0.9785 | 0.9785 | 0.9785 |
| | | Mcc | 0.936 | 0.936 | 0.936 | 0.936 | 0.9391 | 0.9393 | 0.9391 |
| | | obj | −46 | −45.95 | −45.99 | −45.78 | −45.62 | −45.87 | −45.88 |
| | | iteration | 212 | 443 | 329 | 184 | 146 | 160 | 181 |
| diabetes | 1 | AUC | 0.8388 | 0.8393 | 0.839 | 0.8388 | 0.8383 | 0.8378 | 0.8385 |
| | | Accuracy | 0.776 | 0.7747 | 0.7734 | 0.7747 | 0.7708 | 0.7721 | 0.7721 |
| | | Precision | 0.7918 | 0.7904 | 0.789 | 0.7904 | 0.7893 | 0.7886 | 0.7897 |
| | | Recall | 0.89 | 0.89 | 0.89 | 0.89 | 0.884 | 0.888 | 0.886 |
| | | F1 | 0.838 | 0.8373 | 0.8365 | 0.8373 | 0.834 | 0.8354 | 0.8351 |
| | | Mcc | 0.4878 | 0.4846 | 0.4813 | 0.4846 | 0.4759 | 0.4784 | 0.4788 |
| | | obj | −403.1 | −403.03 | −402.97 | −403 | −402.53 | −402.44 | −402.58 |
| | | iteration | 680 | 873 | 612 | 590 | 460 | 475 | 502 |
| | 0 | AUC | 0.8388 | 0.8392 | 0.8391 | 0.8393 | 0.8384 | 0.8383 | 0.839 |
| | | Accuracy | 0.776 | 0.7747 | 0.776 | 0.7747 | 0.7695 | 0.7721 | 0.776 |
| | | Precision | 0.7918 | 0.7904 | 0.7918 | 0.7914 | 0.7858 | 0.7897 | 0.7918 |
| | | Recall | 0.89 | 0.89 | 0.89 | 0.888 | 0.888 | 0.886 | 0.89 |
| | | F1 | 0.838 | 0.8373 | 0.838 | 0.8369 | 0.8338 | 0.8351 | 0.838 |
| | | Mcc | 0.4878 | 0.4846 | 0.4878 | 0.48449 | 0.4718 | 0.4788 | 0.4878 |
| | | obj | −403.1 | −403 | −403.03 | −403 | −402.05 | −402.8 | −402.72 |
| | | iteration | 680 | 793 | 687 | 529 | 490 | 502 | 516 |

| Dataset | Shrinking | Metrics | WSS2 | DPWSS | | | | | |
|---|---|---|---|---|---|---|---|---|---|
| | | | | epsi = 1 sigm = 0.1 | epsi = 1 sigm = 0.3 | epsi = 1 sigm = 0.5 | epsi = 1 sigm = 0.7 | epsi = 0.5 sigm = 0.7 | epsi = 0.1 sigm = 0.7 |
| fourclass | 1 | AUC | 0.8266 | 0.8268 | 0.8262 | 0.8265 | 0.8255 | 0.8251 | 0.8261 |
| | | Accuracy | 0.7715 | 0.7715 | 0.7715 | 0.7715 | 0.7715 | 0.7726 | 0.7726 |
| | | Precision | 0.7455 | 0.7455 | 0.7455 | 0.7477 | 0.7477 | 0.7489 | 0.7489 |
| | | Recall | 0.544 | 0.544 | 0.544 | 0.5407 | 0.5407 | 0.544 | 0.544 |
| | | F1 | 0.629 | 0.629 | 0.629 | 0.6276 | 0.6276 | 0.6302 | 0.6302 |
| | | Mcc | 0.4818 | 0.4818 | 0.4818 | 0.4816 | 0.4816 | 0.4845 | 0.4845 |
| | | obj | −454.29 | −454.27 | −454.22 | −454.23 | −454.12 | −454.12 | −454.17 |
| | | iteration | 590 | 917 | 676 | 509 | 450 | 466 | 496 |
| | 0 | AUC | 0.8266 | 0.827 | 0.8256 | 0.8272 | 0.8245 | 0.8263 | 0.8257 |
| | | Accuracy | 0.7715 | 0.7691 | 0.7715 | 0.7691 | 0.7749 | 0.7726 | 0.7726 |
| | | Precision | 0.7455 | 0.7432 | 0.7455 | 0.7432 | 0.7653 | 0.7534 | 0.7467 |
| | | Recall | 0.544 | 0.5375 | 0.544 | 0.5375 | 0.5309 | 0.5375 | 0.5472 |
| | | F1 | 0.629 | 0.6238 | 0.629 | 0.6238 | 0.6269 | 0.6274 | 0.6316 |
| | | Mcc | 0.4818 | 0.4761 | 0.4818 | 0.4761 | 0.4894 | 0.4842 | 0.4847 |
| | | obj | −454.29 | −454.25 | −454.22 | −454.18 | −453.68 | −453.8 | −454.19 |
| | | iteration | 590 | 908 | 625 | 526 | 421 | 458 | 471 |
| German | 1 | AUC | 0.8165 | 0.8163 | 0.8161 | 0.816 | 0.8161 | 0.8157 | 0.816 |
| | | Accuracy | 0.789 | 0.788 | 0.787 | 0.786 | 0.785 | 0.783 | 0.784 |
| | | Precision | 0.6943 | 0.6947 | 0.69 | 0.6886 | 0.6856 | 0.6861 | 0.6858 |
| | | Recall | 0.53 | 0.5233 | 0.5267 | 0.5233 | 0.5233 | 0.51 | 0.5167 |
| | | F1 | 0.6011 | 0.597 | 0.5974 | 0.5947 | 0.5936 | 0.5851 | 0.5894 |
| | | Mcc | 0.469 | 0.4654 | 0.4638 | 0.4608 | 0.4586 | 0.4514 | 0.455 |
| | | obj | −519.05 | −518.76 | −518.81 | −518.48 | −517.51 | −517.92 | −517.59 |
| | | iteration | 13,688 | 9,415 | 5,821 | 4,533 | 3,675 | 3,741 | 3,787 |
| | 0 | AUC | 0.8165 | 0.8163 | 0.8163 | 0.8155 | 0.8154 | 0.8159 | 0.8162 |
| | | Accuracy | 0.789 | 0.787 | 0.789 | 0.783 | 0.785 | 0.784 | 0.787 |
| | | Precision | 0.6943 | 0.6916 | 0.6978 | 0.6781 | 0.6856 | 0.6826 | 0.6933 |
| | | Recall | 0.53 | 0.5233 | 0.5233 | 0.5267 | 0.5233 | 0.5233 | 0.52 |
| | | F1 | 0.6011 | 0.5958 | 0.5981 | 0.5929 | 0.5936 | 0.5925 | 0.5943 |
| | | Mcc | 0.469 | 0.4631 | 0.4677 | 0.4548 | 0.4586 | 0.4563 | 0.4625 |
| | | obj | −519.05 | −518.64 | −518.69 | −518.41 | −517.91 | −517.98 | −517.65 |
| | | iteration | 13,454 | 9,367 | 5,495 | 4,576 | 3,663 | 3,842 | 3,742 |
| gisette | 1 | AUC | 1 | 1 | 1 | 1 | 1 | 1 | 1 |
| | | Accuracy | 1 | 1 | 1 | 1 | 1 | 1 | 1 |
| | | Precision | 1 | 1 | 1 | 1 | 1 | 1 | 1 |
| | | Recall | 1 | 1 | 1 | 1 | 1 | 1 | 1 |
| | | F1 | 1 | 1 | 1 | 1 | 1 | 1 | 1 |
| | | Mcc | 1 | 1 | 1 | 1 | 1 | 1 | 1 |
| | | obj | −0.668 | −0.668 | −0.668 | −0.668 | −0.668 | −0.668 | −0.668 |
| | | iteration | 8,157 | 23,247 | 9,312 | 6,736 | 6,002 | 5,978 | 5,979 |

(Continued)

| Dataset | Shrinking | Metrics | WSS2 | DPWSS epsi = 1 sigm = 0.1 | epsi = 1 sigm = 0.3 | epsi = 1 sigm = 0.5 | epsi = 1 sigm = 0.7 | epsi = 0.5 sigm = 0.7 | epsi = 0.1 sigm = 0.7 |
|---|---|---|---|---|---|---|---|---|---|
| | 0 | AUC | 1 | 1 | 1 | 1 | 1 | 1 | 1 |
| | | Accuracy | 1 | 1 | 1 | 1 | 1 | 1 | 1 |
| | | Precision | 1 | 1 | 1 | 1 | 1 | 1 | 1 |
| | | Recall | 1 | 1 | 1 | 1 | 1 | 1 | 1 |
| | | F1 | 1 | 1 | 1 | 1 | 1 | 1 | 1 |
| | | Mcc | 1 | 1 | 1 | 1 | 1 | 1 | 1 |
| | | obj | −0.668 | −0.668 | −0.668 | −0.668 | −0.668 | −0.668 | −0.668 |
| | | iteration | 8,246 | 17,902 | 7,933 | 6,918 | 6,258 | 6,225 | 6,250 |
| heart | 1 | AUC | 0.9282 | 0.9281 | 0.9287 | 0.9296 | 0.9287 | 0.9282 | 0.9296 |
| | | Accuracy | 0.8481 | 0.8444 | 0.8481 | 0.8481 | 0.8519 | 0.8593 | 0.8481 |
| | | Precision | 0.8376 | 0.8362 | 0.8376 | 0.8376 | 0.8509 | 0.8534 | 0.8376 |
| | | Recall | 0.8167 | 0.8083 | 0.8167 | 0.8167 | 0.8083 | 0.825 | 0.8167 |
| | | F1 | 0.827 | 0.822 | 0.827 | 0.827 | 0.8291 | 0.839 | 0.827 |
| | | Mcc | 0.6919 | 0.6843 | 0.6919 | 0.6919 | 0.6992 | 0.7144 | 0.6919 |
| | | obj | −92.47 | −92.07 | −92.33 | −92.17 | −90.78 | −91.34 | −91.62 |
| | | iteration | 1,010 | 992 | 662 | 525 | 372 | 404 | 410 |
| | 0 | AUC | 0.9282 | 0.9278 | 0.9284 | 0.9292 | 0.9251 | 0.9278 | 0.9275 |
| | | Accuracy | 0.8481 | 0.8481 | 0.8444 | 0.8556 | 0.8556 | 0.8593 | 0.8481 |
| | | Precision | 0.8376 | 0.8435 | 0.8362 | 0.8462 | 0.8584 | 0.8596 | 0.8376 |
| | | Recall | 0.8167 | 0.8083 | 0.8083 | 0.825 | 0.8083 | 0.8167 | 0.8167 |
| | | F1 | 0.827 | 0.8255 | 0.822 | 0.8354 | 0.8326 | 0.8376 | 0.827 |
| | | Mcc | 0.6919 | 0.6917 | 0.6843 | 0.7069 | 0.7068 | 0.7143 | 0.6919 |
| | | obj | −92.47 | −92.2 | −92.07 | −92.09 | −91.03 | −91.14 | −91.36 |
| | | iteration | 1,010 | 1,097 | 671 | 542 | 380 | 390 | 411 |
| ijcnn1 | 1 | AUC | 0.918 | 0.918 | 0.918 | 0.9179 | 0.917 | 0.9184 | 0.9179 |
| | | Accuracy | 0.9242 | 0.9242 | 0.9241 | 0.9242 | 0.9241 | 0.9241 | 0.9241 |
| | | Precision | 0.7579 | 0.758 | 0.7576 | 0.7581 | 0.767 | 0.7565 | 0.7598 |
| | | Recall | 0.3219 | 0.3215 | 0.3215 | 0.3217 | 0.314 | 0.3221 | 0.3188 |
| | | F1 | 0.4518 | 0.4515 | 0.4514 | 0.4517 | 0.4456 | 0.4518 | 0.4491 |
| | | Mcc | 0.4628 | 0.4626 | 0.4624 | 0.4628 | 0.4604 | 0.4625 | 0.4612 |
| | | obj | -8,590.16 | -8,590.07 | -8,590.11 | -8,590.07 | -8,588.99 | -8,588.82 | -8,589 |
| | | iteration | 18,382 | 40,443 | 29,635 | 25,599 | 19,150 | 18,694 | 17,842 |
| | 0 | AUC | 0.918 | 0.9181 | 0.918 | 0.9181 | 0.9181 | 0.9183 | 0.9183 |
| | | Accuracy | 0.9241 | 0.924 | 0.9241 | 0.9241 | 0.9238 | 0.9242 | 0.924 |
| | | Precision | 0.7574 | 0.7573 | 0.7573 | 0.7572 | 0.7549 | 0.7569 | 0.7562 |
| | | Recall | 0.3217 | 0.3202 | 0.3215 | 0.3212 | 0.3179 | 0.3221 | 0.3208 |
| | | F1 | 0.4515 | 0.4501 | 0.4513 | 0.4511 | 0.4474 | 0.4519 | 0.4505 |
| | | Mcc | 0.4625 | 0.4614 | 0.4623 | 0.4621 | 0.4588 | 0.4626 | 0.4614 |
| | | obj | −8,590.16 | −8,590.05 | −8,590.1 | −8,589.94 | −8,588.84 | −8,589.73 | −8,589.18 |
| | | iteration | 16,469 | 4,5191 | 30,133 | 23,786 | 18,416 | 20,637 | 18,750 |

| Dataset | Shrinking | Metrics | WSS2 | DPWSS epsi = 1 sigm = 0.1 | epsi = 1 sigm = 0.3 | epsi = 1 sigm = 0.5 | epsi = 1 sigm = 0.7 | epsi = 0.5 sigm = 0.7 | epsi = 0.1 sigm = 0.7 |
|---|---|---|---|---|---|---|---|---|---|
| ionosphere | 1 | AUC | 0.9677 | 0.9684 | 0.9681 | 0.9679 | 0.9686 | 0.9688 | 0.9687 |
| | | Accuracy | 0.9373 | 0.9373 | 0.9373 | 0.9259 | 0.9345 | 0.9373 | 0.9345 |
| | | Precision | 0.9283 | 0.9283 | 0.9283 | 0.9234 | 0.9244 | 0.9356 | 0.9316 |
| | | Recall | 0.9778 | 0.9778 | 0.9778 | 0.9644 | 0.9778 | 0.9689 | 0.9689 |
| | | F1 | 0.9524 | 0.9524 | 0.9524 | 0.9435 | 0.9503 | 0.952 | 0.9499 |
| | | Mcc | 0.8634 | 0.8634 | 0.8634 | 0.8379 | 0.8572 | 0.863 | 0.8567 |
| | | obj | −73.41 | −73.41 | −73.31 | −71.97 | −72.02 | −72.44 | −72.11 |
| | | iteration | 1,016 | 1,489 | 834 | 664 | 555 | 562 | 525 |
| | 0 | AUC | 0.9677 | 0.9674 | 0.9678 | 0.9673 | 0.9651 | 0.9667 | 0.965 |
| | | Accuracy | 0.9373 | 0.9288 | 0.9373 | 0.9288 | 0.9316 | 0.9288 | 0.9288 |
| | | Precision | 0.9283 | 0.9274 | 0.9283 | 0.9237 | 0.9277 | 0.9237 | 0.9274 |
| | | Recall | 0.9778 | 0.9644 | 0.9778 | 0.9689 | 0.9689 | 0.9689 | 0.9644 |
| | | F1 | 0.9524 | 0.9455 | 0.9524 | 0.9458 | 0.9478 | 0.9458 | 0.9455 |
| | | Mcc | 0.8634 | 0.8441 | 0.8634 | 0.8442 | 0.8505 | 0.8442 | 0.8441 |
| | | obj | −73.41 | −73.15 | −73.2 | −73.13 | −72.44 | −72.85 | −71.91 |
| | | iteration | 770 | 1,348 | 944 | 761 | 560 | 610 | 548 |
| rcv1 | 1 | AUC | 0.9989 | 0.9989 | 0.9989 | 0.9989 | 0.9989 | 0.9989 | 0.9989 |
| | | Accuracy | 0.9896 | 0.9896 | 0.9896 | 0.9896 | 0.9896 | 0.9896 | 0.9896 |
| | | Precision | 0.9896 | 0.9896 | 0.9896 | 0.9896 | 0.9896 | 0.9896 | 0.9897 |
| | | Recall | 0.9903 | 0.9903 | 0.9903 | 0.9903 | 0.9903 | 0.9903 | 0.9903 |
| | | F1 | 0.9899 | 0.9899 | 0.9899 | 0.9899 | 0.9899 | 0.9899 | 0.99 |
| | | Mcc | 0.9791 | 0.9791 | 0.9791 | 0.9791 | 0.9791 | 0.9791 | 0.9792 |
| | | obj | −1,745.67 | −1,745.66 | −1,745.65 | −1,745.62 | −1,745.63 | −1,745.6 | −1,745.59 |
| | | iteration | 11,639 | 41,681 | 17,129 | 12,374 | 11,029 | 9,865 | 9,945 |
| | 0 | AUC | 0.9989 | 0.9989 | 0.9989 | 0.9989 | 0.9989 | 0.9989 | 0.9989 |
| | | Accuracy | 0.9896 | 0.9896 | 0.9896 | 0.9896 | 0.9896 | 0.9896 | 0.9896 |
| | | Precision | 0.9896 | 0.9896 | 0.9896 | 0.9896 | 0.9896 | 0.9897 | 0.9896 |
| | | Recall | 0.9903 | 0.9903 | 0.9903 | 0.9903 | 0.9903 | 0.9903 | 0.9903 |
| | | F1 | 0.9899 | 0.9899 | 0.9899 | 0.9899 | 0.9899 | 0.99 | 0.9899 |
| | | Mcc | 0.9791 | 0.9791 | 0.9791 | 0.9791 | 0.9791 | 0.9792 | 0.9791 |
| | | obj | −1,745.67 | −1,745.66 | −1,745.65 | −1,745.64 | −1745.55 | −1,745.58 | −1,745.61 |
| | | iteration | 11,650 | 38,114 | 16,388 | 13,014 | 9,242 | 9,419 | 10,457 |
| sonar | 1 | AUC | 0.9495 | 0.9491 | 0.9475 | 0.9467 | 0.9489 | 0.9459 | 0.9482 |
| | | Accuracy | 0.8942 | 0.8894 | 0.8894 | 0.8894 | 0.8942 | 0.899 | 0.8894 |
| | | Precision | 0.8641 | 0.8558 | 0.8558 | 0.8558 | 0.8641 | 0.8654 | 0.8558 |
| | | Recall | 0.9175 | 0.9175 | 0.9175 | 0.9175 | 0.9175 | 0.9278 | 0.9175 |
| | | F1 | 0.89 | 0.8856 | 0.8856 | 0.8856 | 0.89 | 0.8955 | 0.8856 |
| | | Mcc | 0.7896 | 0.7806 | 0.7806 | 0.7806 | 0.7896 | 0.7999 | 0.7806 |
| | | obj | −65.67 | −65.62 | −65.48 | −65.49 | −65.21 | −64.79 | −65 |
| | | iteration | 1,492 | 1,716 | 1,035 | 687 | 544 | 478 | 473 |

(Continued)

| Table 3 (continued) | | | | | | | | | |
|---|---|---|---|---|---|---|---|---|---|
| **Dataset** | **Shrinking** | **Metrics** | **WSS2** | **DPWSS** | | | | | |
| | | | | **epsi = 1 sigm = 0.1** | **epsi = 1 sigm = 0.3** | **epsi = 1 sigm = 0.5** | **epsi = 1 sigm = 0.7** | **epsi = 0.5 sigm = 0.7** | **epsi = 0.1 sigm = 0.7** |
| | 0 | AUC | 0.9496 | 0.9501 | 0.9475 | 0.9464 | 0.9492 | 0.9455 | 0.9439 |
| | | Accuracy | 0.8942 | 0.8894 | 0.8894 | 0.8894 | 0.899 | 0.899 | 0.8846 |
| | | Precision | 0.8641 | 0.8558 | 0.8558 | 0.8558 | 0.8654 | 0.8585 | 0.8476 |
| | | Recall | 0.9175 | 0.9175 | 0.9175 | 0.9175 | 0.9278 | 0.9381 | 0.9175 |
| | | F1 | 0.89 | 0.8856 | 0.8856 | 0.8856 | 0.8955 | 0.8966 | 0.8812 |
| | | Mcc | 0.7896 | 0.7806 | 0.7806 | 0.7806 | 0.7999 | 0.8013 | 0.7717 |
| | | obj | −65.67 | −65.57 | −65.43 | −65.49 | −64.97 | −65.02 | −64.88 |
| | | iteration | 1,397 | 1,516 | 929 | 701 | 489 | 448 | 440 |
| splice | 1 | AUC | 0.9173 | 0.9165 | 0.9164 | 0.9169 | 0.9165 | 0.9173 | 0.9162 |
| | | Accuracy | 0.842 | 0.84 | 0.842 | 0.845 | 0.841 | 0.84 | 0.844 |
| | | Precision | 0.8671 | 0.8665 | 0.8716 | 0.8724 | 0.8698 | 0.868 | 0.8737 |
| | | Recall | 0.8201 | 0.8162 | 0.8143 | 0.8201 | 0.8143 | 0.8143 | 0.8162 |
| | | F1 | 0.8429 | 0.8406 | 0.842 | 0.8455 | 0.8412 | 0.8403 | 0.844 |
| | | Mcc | 0.6853 | 0.6815 | 0.6859 | 0.6916 | 0.6838 | 0.6817 | 0.69 |
| | | obj | −375.19 | −374.55 | −374.56 | −374.31 | −373.25 | −374.02 | −373.51 |
| | | iteration | 95,972 | 19,779 | 11,627 | 8,079 | 6,486 | 6,847 | 6,620 |
| | 0 | AUC | 0.9173 | 0.9164 | 0.9169 | 0.9156 | 0.9166 | 0.9163 | 0.9168 |
| | | Accuracy | 0.842 | 0.845 | 0.843 | 0.843 | 0.841 | 0.844 | 0.841 |
| | | Precision | 0.8671 | 0.8724 | 0.8719 | 0.8719 | 0.8698 | 0.8737 | 0.8698 |
| | | Recall | 0.8201 | 0.8201 | 0.8162 | 0.8162 | 0.8143 | 0.8162 | 0.8143 |
| | | F1 | 0.8429 | 0.8455 | 0.8432 | 0.8432 | 0.8412 | 0.844 | 0.8412 |
| | | Mcc | 0.6853 | 0.6916 | 0.6878 | 0.6878 | 0.6838 | 0.69 | 0.6838 |
| | | obj | −375.19 | −374.08 | −374.11 | −373.89 | −373.32 | −373.77 | −373.7 |
| | | iteration | 38,987 | 18,752 | 11,058 | 7,562 | 6,501 | 6,845 | 6,785 |
| w1a | 1 | AUC | 0.9755 | 0.9759 | 0.9757 | 0.9757 | 0.9757 | 0.9756 | 0.9754 |
| | | Accuracy | 0.9927 | 0.9927 | 0.9927 | 0.9927 | 0.9927 | 0.9927 | 0.9927 |
| | | Precision | 0.9821 | 0.9821 | 0.9821 | 0.9821 | 0.9821 | 0.9821 | 0.9821 |
| | | Recall | 0.7639 | 0.7639 | 0.7639 | 0.7639 | 0.7639 | 0.7639 | 0.7639 |
| | | F1 | 0.8594 | 0.8594 | 0.8594 | 0.8594 | 0.8594 | 0.8594 | 0.8594 |
| | | Mcc | 0.8628 | 0.8628 | 0.8628 | 0.8628 | 0.8628 | 0.8628 | 0.8628 |
| | | obj | −62.92 | −62.89 | −62.91 | −62.9 | −62.85 | −62.89 | −62.89 |
| | | iteration | 2,565 | 9,030 | 5,034 | 3,529 | 1,835 | 1,949 | 2,224 |
| | 0 | AUC | 0.9755 | 0.9755 | 0.9758 | 0.9759 | 0.9766 | 0.9758 | 0.9734 |
| | | Accuracy | 0.9927 | 0.9927 | 0.9927 | 0.9927 | 0.9927 | 0.9927 | 0.9927 |
| | | Precision | 0.9821 | 0.9821 | 0.9821 | 0.9821 | 0.9821 | 0.9821 | 0.9821 |
| | | Recall | 0.7639 | 0.7639 | 0.7639 | 0.7639 | 0.7639 | 0.7639 | 0.7639 |
| | | F1 | 0.8594 | 0.8594 | 0.8594 | 0.8594 | 0.8594 | 0.8594 | 0.8594 |
| | | Mcc | 0.8628 | 0.8628 | 0.8628 | 0.8628 | 0.8628 | 0.8628 | 0.8628 |
| | | obj | −62.92 | −62.89 | −62.89 | −62.79 | −62.8 | −62.86 | −62.85 |
| | | iteration | 2,547 | 8,436 | 4,326 | 2,222 | 1,691 | 1,781 | 1,737 |

| Table 3 (continued) | | | | | | | | | |
|---|---|---|---|---|---|---|---|---|---|
| Dataset | Shrinking | Metrics | WSS2 | DPWSS | | | | | |
| | | | | epsi = 1 sigm = 0.1 | epsi = 1 sigm = 0.3 | epsi = 1 sigm = 0.5 | epsi = 1 sigm = 0.7 | epsi = 0.5 sigm = 0.7 | epsi = 0.1 sigm = 0.7 |
| w5a | 1 | AUC | 0.9632 | 0.9632 | 0.9638 | 0.9632 | 0.9625 | 0.9653 | 0.9626 |
| | | Accuracy | 0.9889 | 0.9887 | 0.9888 | 0.9887 | 0.9887 | 0.9889 | 0.9887 |
| | | Precision | 0.9524 | 0.9424 | 0.9474 | 0.9424 | 0.9424 | 0.9524 | 0.9424 |
| | | Recall | 0.6406 | 0.6406 | 0.6406 | 0.6406 | 0.6406 | 0.6406 | 0.6406 |
| | | F1 | 0.766 | 0.7627 | 0.7643 | 0.7627 | 0.7627 | 0.766 | 0.7627 |
| | | Mcc | 0.7762 | 0.772 | 0.7741 | 0.772 | 0.772 | 0.7762 | 0.772 |
| | | obj | −291.68 | −291.55 | −291.6 | −291.6 | −291.37 | −290.15 | −291.29 |
| | | iteration | 15,422 | 31,105 | 13,388 | 10,568 | 6,514 | 5,906 | 5,858 |
| | 0 | AUC | 0.9632 | 0.9636 | 0.9632 | 0.9633 | 0.9632 | 0.9623 | 0.9626 |
| | | Accuracy | 0.9889 | 0.9888 | 0.9887 | 0.9888 | 0.9889 | 0.9888 | 0.9887 |
| | | Precision | 0.9524 | 0.9427 | 0.9424 | 0.9474 | 0.9476 | 0.9474 | 0.9424 |
| | | Recall | 0.6406 | 0.6441 | 0.6406 | 0.6406 | 0.6441 | 0.6406 | 0.6406 |
| | | F1 | 0.766 | 0.7653 | 0.7627 | 0.7643 | 0.7669 | 0.7643 | 0.7627 |
| | | Mcc | 0.7762 | 0.7743 | 0.772 | 0.7741 | 0.7764 | 0.7741 | 0.772 |
| | | obj | −291.68 | −291.53 | −291.55 | −291.46 | −291.39 | −291.4 | −291.31 |
| | | iteration | 15,511 | 25,969 | 13,074 | 8,879 | 6,630 | 6,373 | 5,838 |

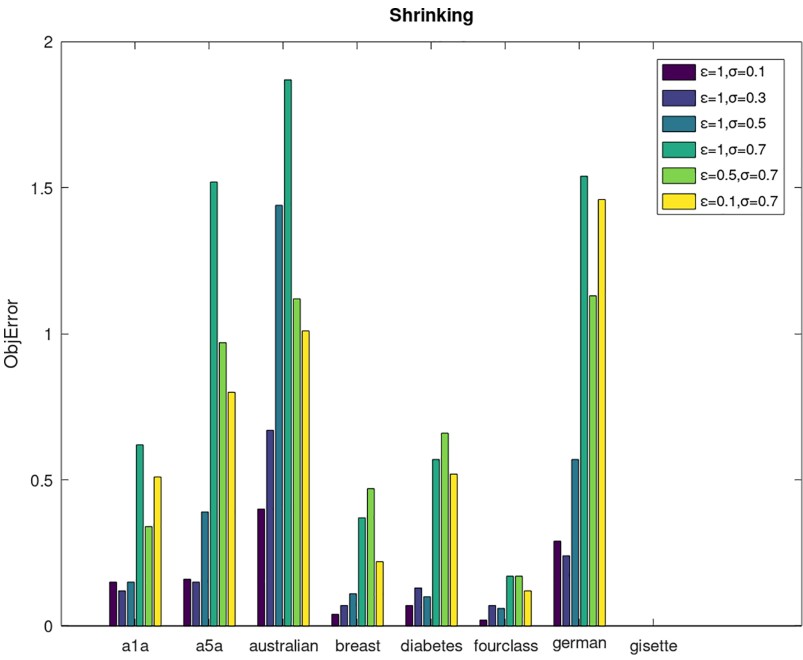

**Figure 2 The algorithm stability of DPWSS for different $\varepsilon$ and $\sigma$ vs WSS 2 on dataset 1 to 8 with shrinking.**

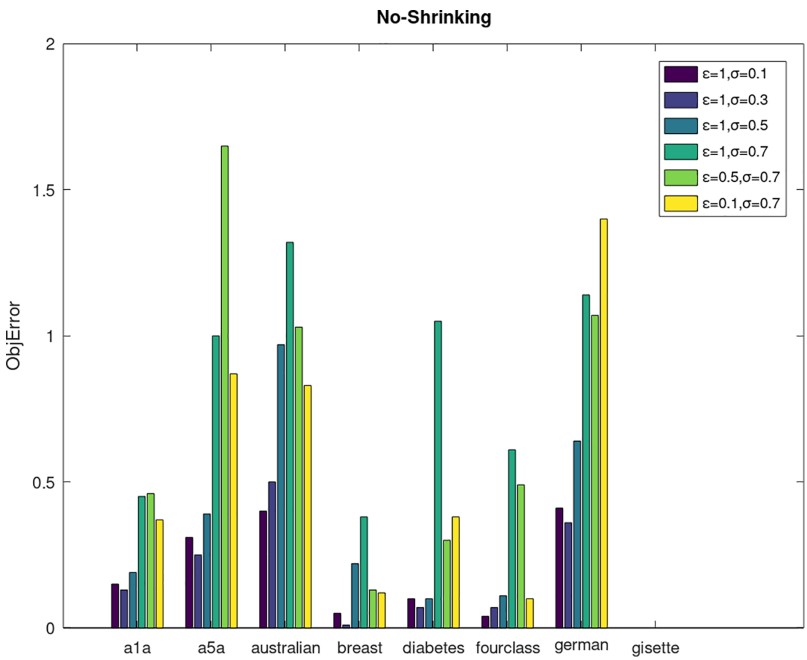

**Figure 3** The algorithm stability of DPWSS for different $\varepsilon$ and $\sigma$ *vs* WSS 2 on dataset 1 to 8 without shrinking.

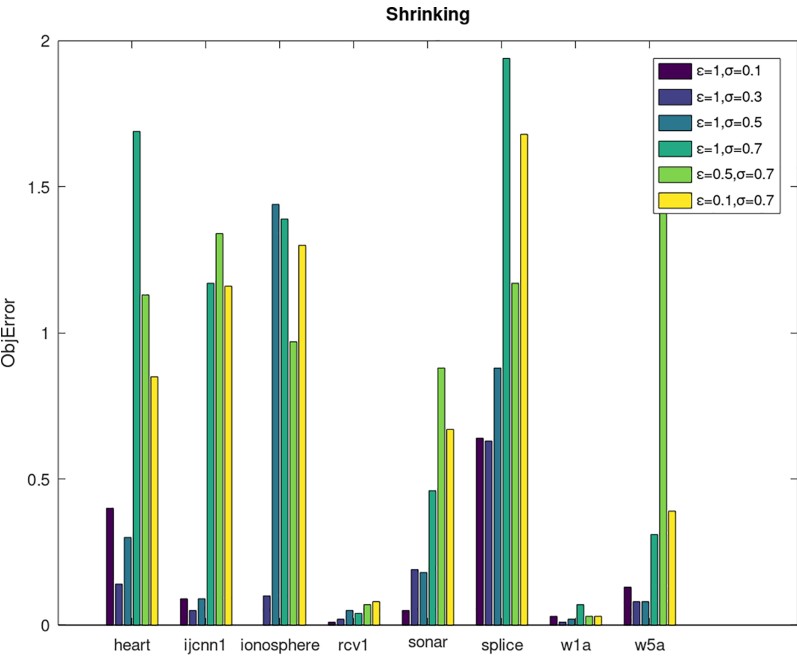

**Figure 4** The algorithm stability of DPWSS for different $\varepsilon$ and $\sigma$ *vs* WSS 2 on dataset 9 to 16 with shrinking.

In the above experiments, we compared the average results of five times running of the DPWSS algorithm with the WSS 2 algorithm. It can be seen from the experimental results that the two algorithms have similar classification capability and optimized objective

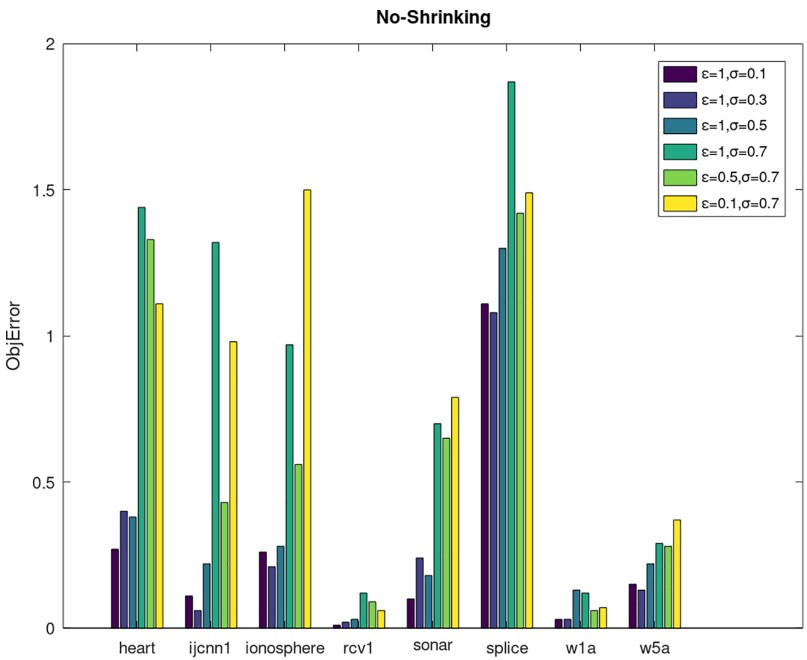

**Figure 5** The algorithm stability of DPWSS for different $\varepsilon$ and $\sigma$ *vs* WSS 2 on dataset 9 to 16 without shrinking.

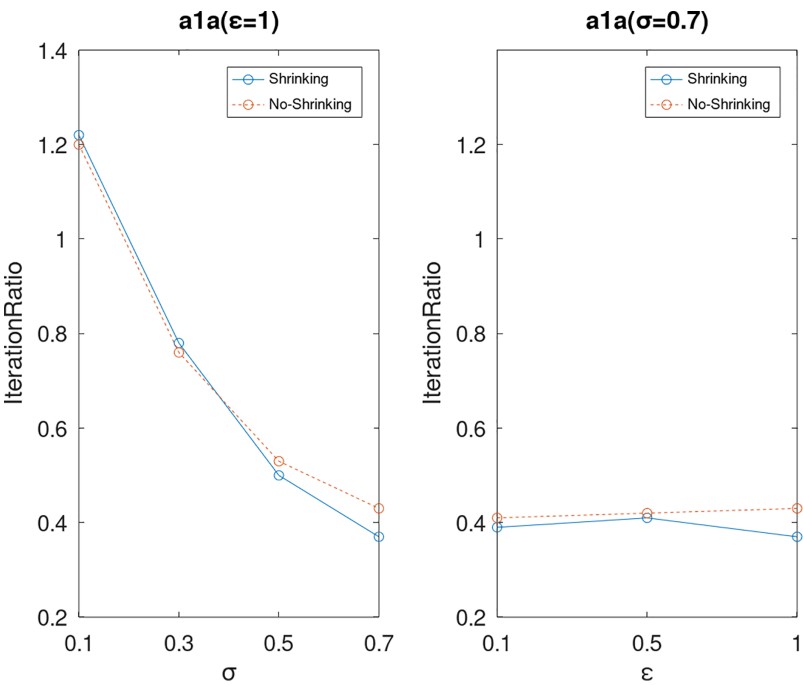

**Figure 6** The execution efficiency of DPWSS for different $\varepsilon$ and $\sigma$ *vs* WSS 2 on dataset a1a.

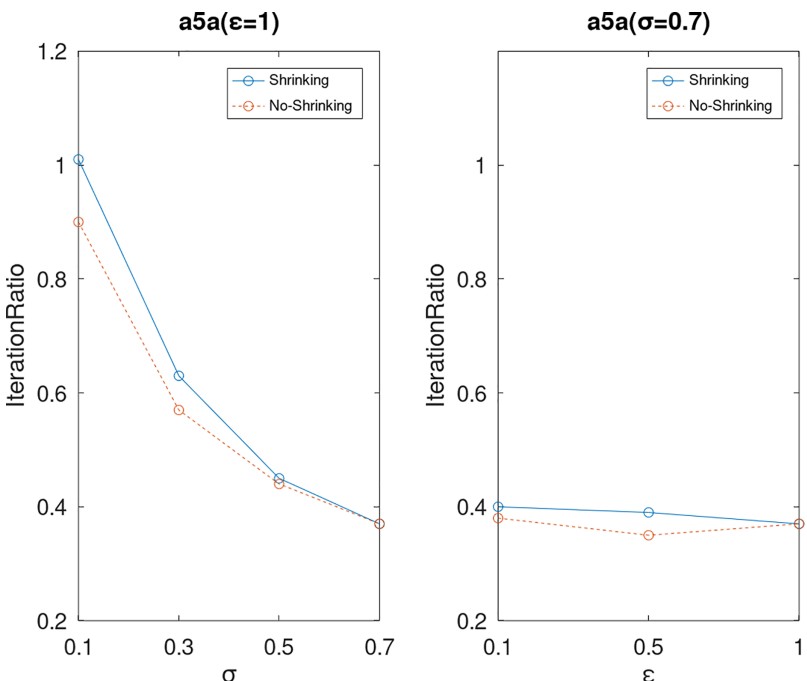

**Figure 7 The execution efficiency of DPWSS for different ε and σ vs WSS 2 on dataset a5a.**

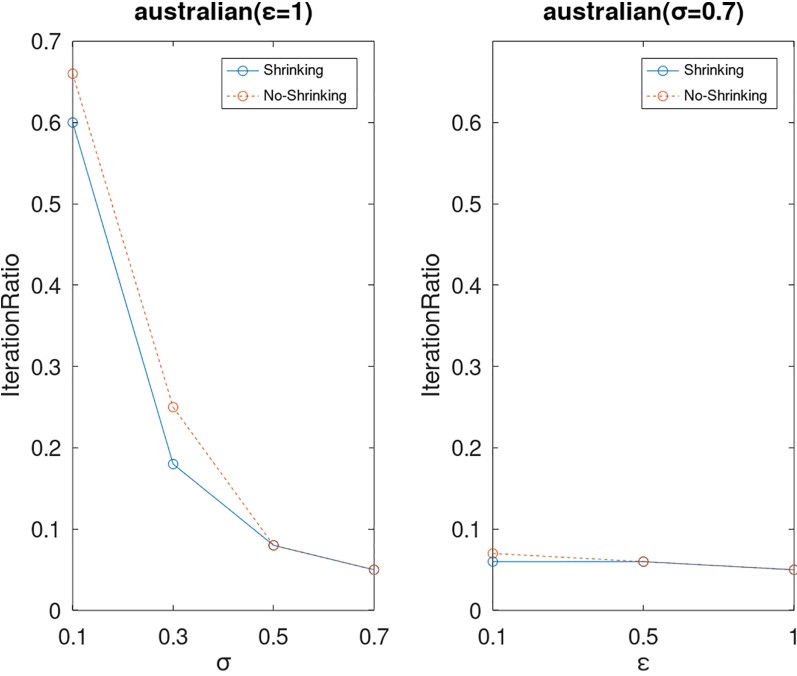

**Figure 8 The execution efficiency of DPWSS for different ε and σ vs WSS 2 on dataset Australian.**

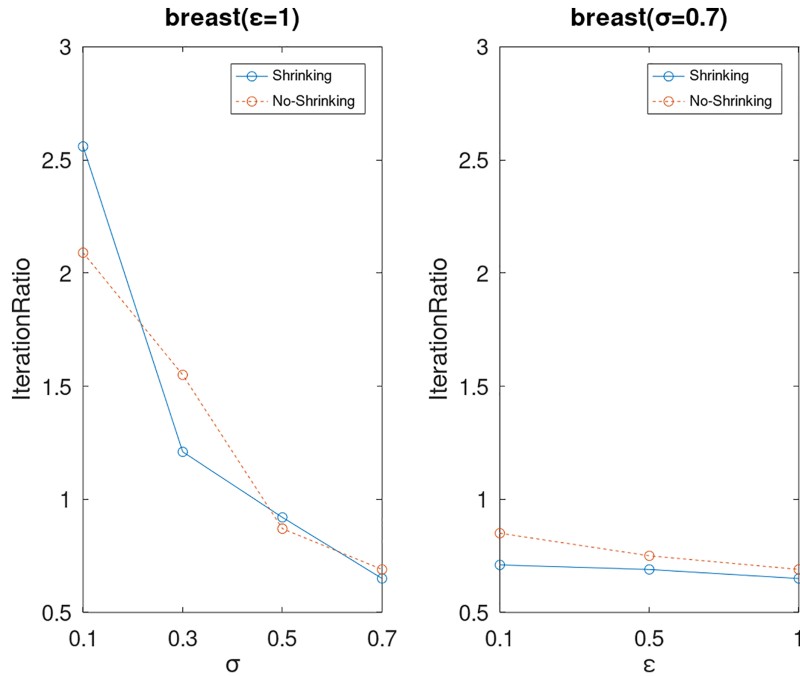

**Figure 9 The execution efficiency of DPWSS for different ε and σ vs WSS 2 on dataset breast.**

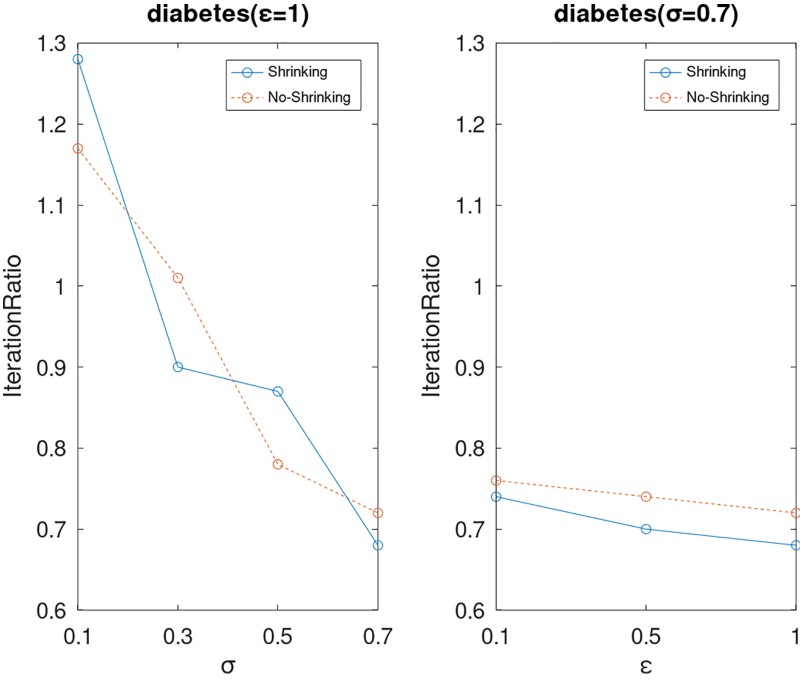

**Figure 10 The execution efficiency of DPWSS for different ε and σ vs WSS 2 on dataset diabetes.**

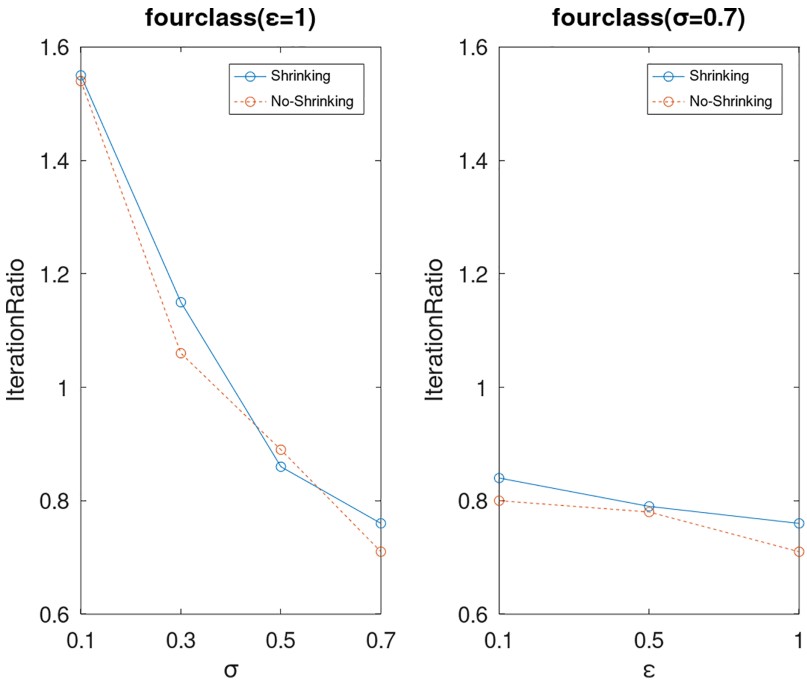

**Figure 11** **The execution efficiency of DPWSS for different ε and σ vs WSS 2 on dataset fourclass.**

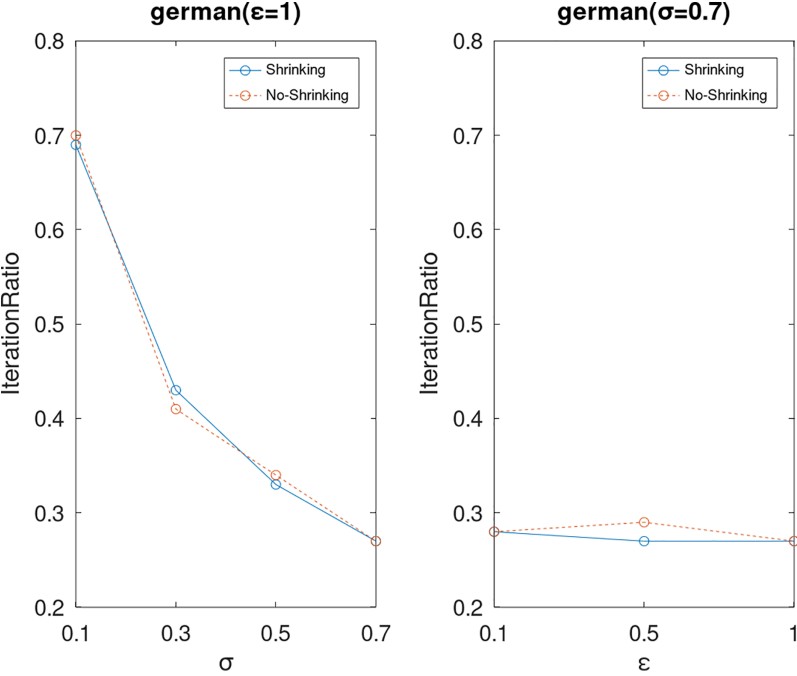

**Figure 12** **The execution efficiency of DPWSS for different ε and σ vs WSS 2 on dataset German.**

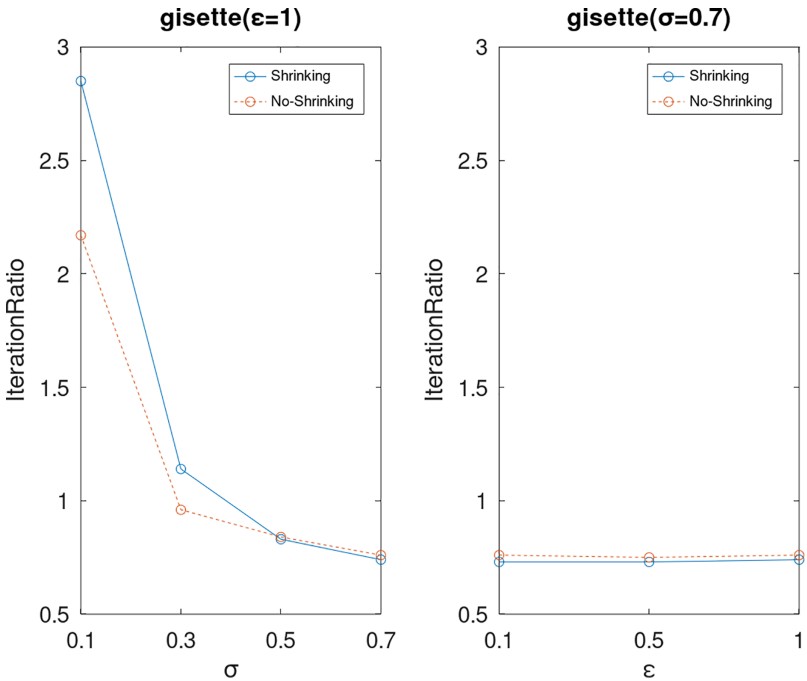

**Figure 13 The execution efficiency of DPWSS for different ε and σ vs WSS 2 on dataset gisette.**

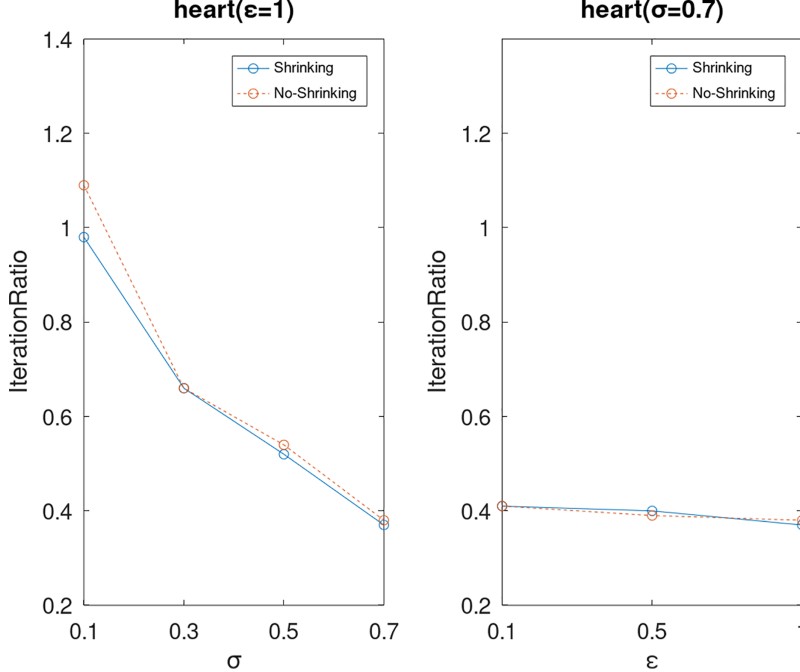

**Figure 14 The execution efficiency of DPWSS for different ε and σ vs WSS 2 on dataset heart.**

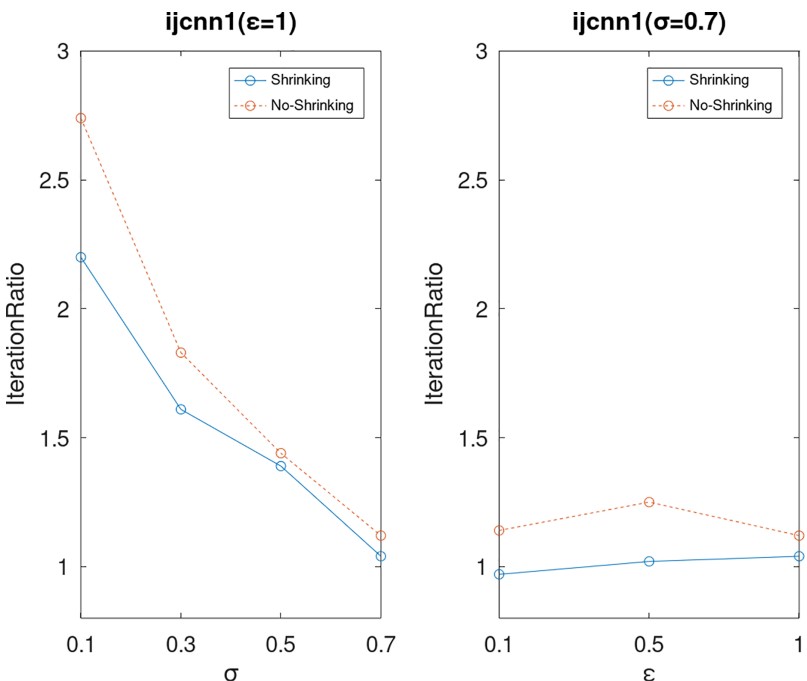

**Figure 15 The execution efficiency of DPWSS for different *ε* and *σ* *vs* WSS 2 on dataset ijcnn1.**

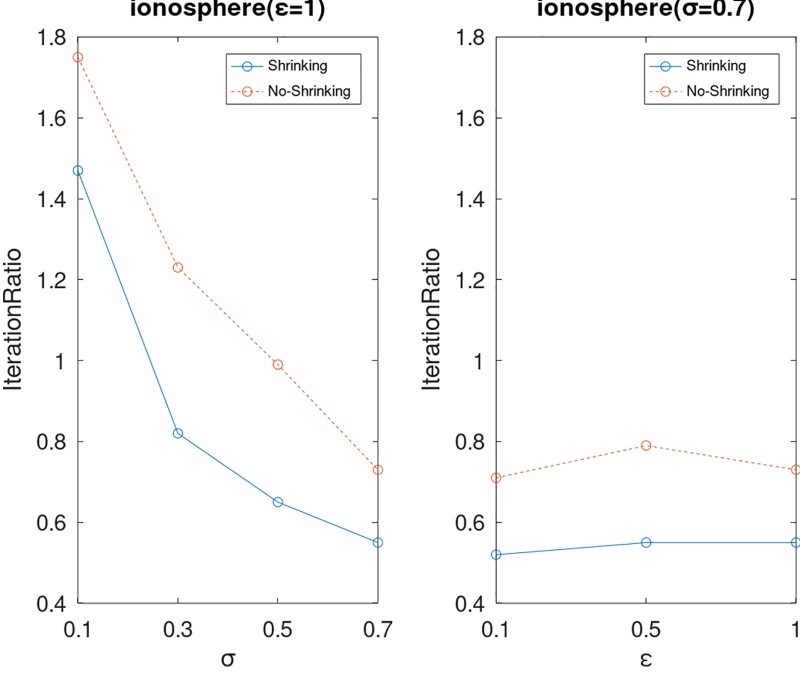

**Figure 16 The execution efficiency of DPWSS for different *ε* and *σ* *vs* WSS 2 on dataset ionosphere.**

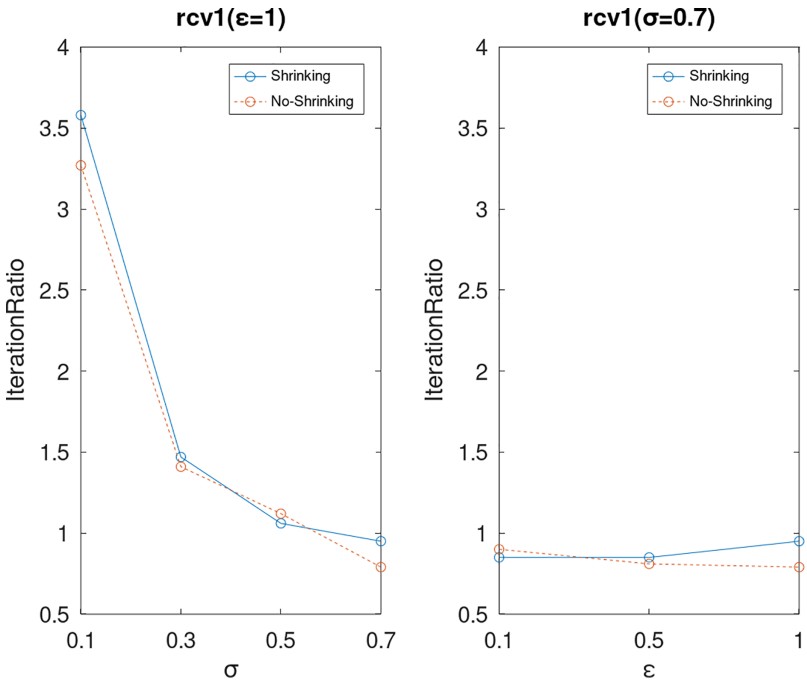

**Figure 17 The execution efficiency of DPWSS for different ε and σ vs WSS 2 on dataset rcv1.**

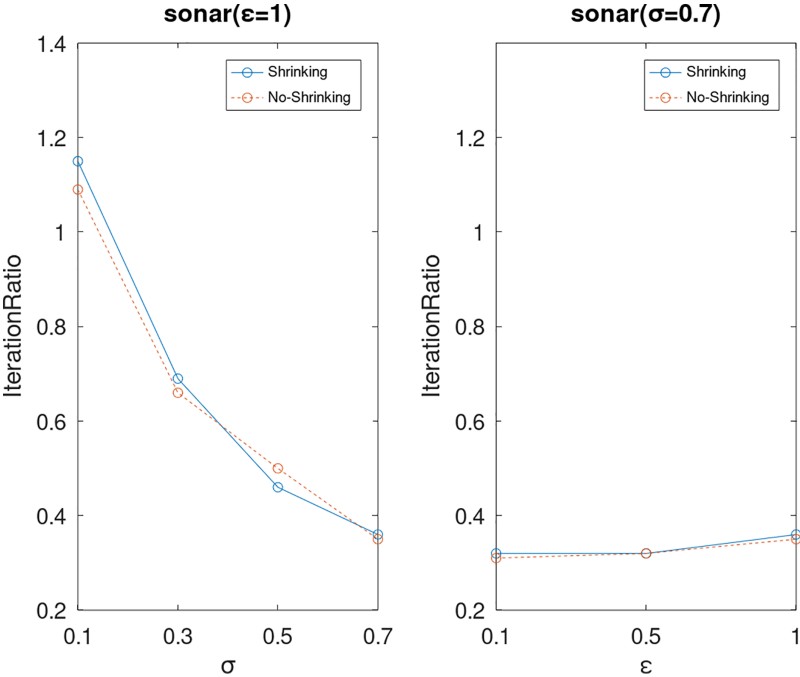

**Figure 18 The execution efficiency of DPWSS for different ε and σ vs WSS 2 on dataset sonar.**

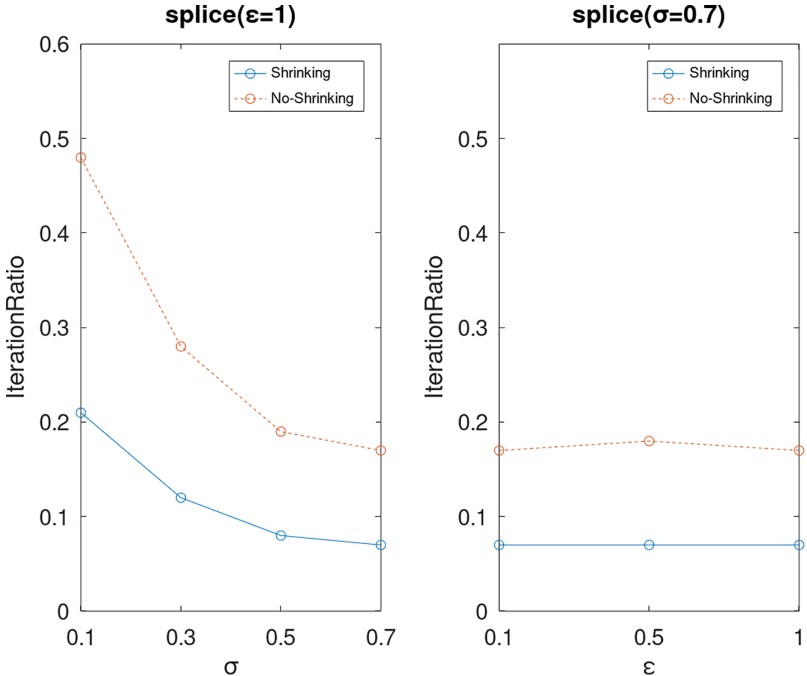

**Figure 19 The execution efficiency of DPWSS for different ε and σ vs WSS 2 on dataset splice.**

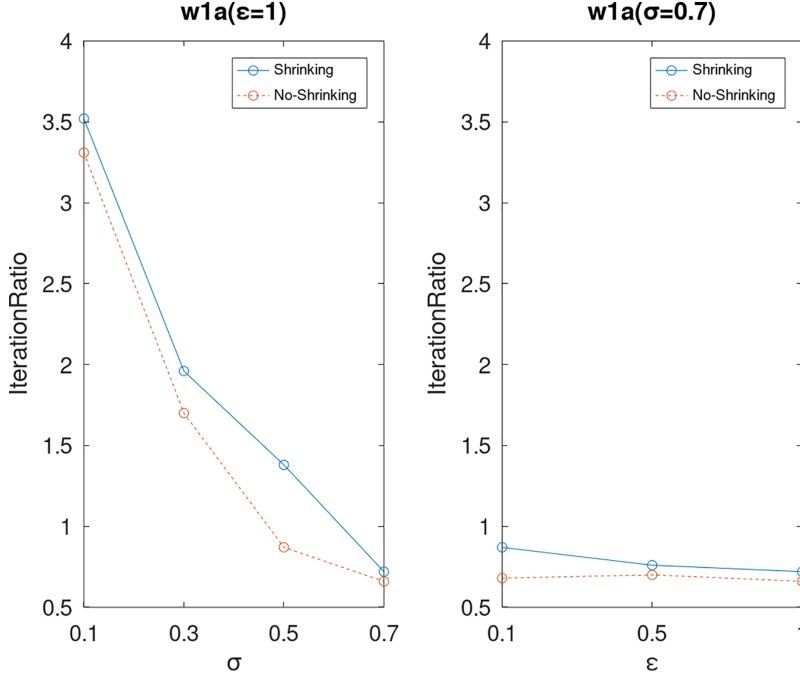

**Figure 20 The execution efficiency of DPWSS for different ε and σ vs WSS 2 on dataset w1a.**

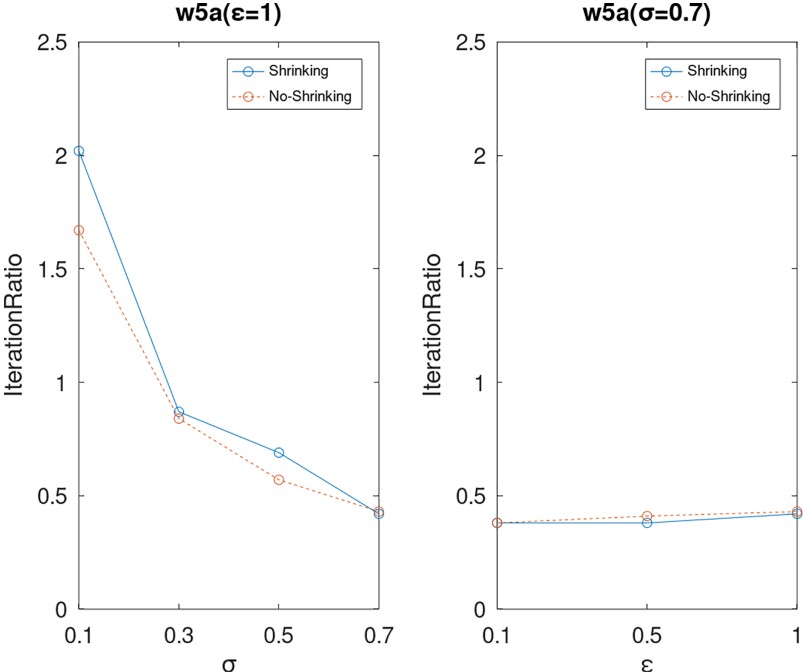

**Figure 21 The execution efficiency of DPWSS for different $\varepsilon$ and $\sigma$ vs WSS 2 on dataset w5a.**

value under different parameter combinations. Under the same set of parameters, the experimental results of DPWSS algorithm differ little each time, and the main difference lies in the iterations. These slight differences show that the DPWSS algorithm has good usability while satisfying DP. Due to the limitations of the paper, we have not listed each running result in the experiments.

## CONCLUSIONS

In this paper, we study the privacy leakage problem of the traditional SVM training methods. The DPWSS algorithm was proposed to release a private classification model of SVM and theoretically proved to satisfy DP through utilizing the exponential mechanism to privately select working sets in every iteration. The extended experiments show that the DPWSS algorithm achieves similar classification capability and the optimized objective value as the original non-privacy SVM under different parameters. Meanwhile, the DPWSS algorithm has a higher execution efficiency by comparing iterations on different datasets. In the DPWSS algorithm, randomness is introduced in the training process. The most prominent advantages include that there are no requirements for differentiability of the objective function and complex sensitivity analysis compared with objective perturbation or output perturbation methods. And a number of training set selection methods can be easily combined with the DPWSS algorithm for large-scale training problems that require large memory and enormous amounts of training time. Because the DPWSS algorithm doesn't change the training process of the classical non-privacy SVMs, it is also suitable for multi-class classification. It is a challenge that

parameter setting of the constant-factor $\sigma$ for different datasets. The idea of introducing randomness into the optimization process can be easily extended to other privacy-preserving machine learning algorithms, and how to ensure that the method meets the DP requirements is another challenge. Furthermore, the DPWSS algorithm is valid to release a private classification model for linear SVM, while invalid for other non-linear kernel SVM as the privacy disclosure problem of the support vectors in kernel function. In future work, we will study how to release a private classification model for non-linear kernel SVMs.

### Funding
This work was supported by the National Natural Science Foundation of China under Grant 61672179, 61370083, 61402126 and 61501275, by the Natural Science Foundation of Heilongjiang Province under Grant F2015030, by the Science Fund for Youths of Heilongjiang Province under Grant QC2016083, by the Postdoctoral Fellowship of Heilongjiang Province under Grant LBH - Z14071, by the Fundamental Research Funds in Heilongjiang Provincial Universities under Grant 135109245. There was no additional external funding received for this study. The funders had no role in study design, data collection and analysis, decision to publish, or preparation of the manuscript.

### Grant Disclosures
The following grant information was disclosed by the authors:
National Natural Science Foundation of China: 61672179, 61370083, 61402126 and 61501275.
Natural Science Foundation of Heilongjiang Province: F2015030.
Science Fund for Youths of Heilongjiang Province: QC2016083.
Postdoctoral Fellowship of Heilongjiang Province: LBH-Z14071.
Fundamental Research Funds in Heilongjiang Provincial Universities: 135109245.

### Competing Interests
The authors declare that they have no competing interests.

### Author Contributions
- Zhenlong Sun conceived and designed the experiments, performed the experiments, analyzed the data, performed the computation work, prepared figures and/or tables, authored or reviewed drafts of the paper, and approved the final draft.
- Jing Yang conceived and designed the experiments, analyzed the data, authored or reviewed drafts of the paper, and approved the final draft.
- Xiaoye Li performed the experiments, performed the computation work, prepared figures and/or tables, and approved the final draft.
- Jianpei Zhang analyzed the data, authored or reviewed drafts of the paper, and approved the final draft.

## Data Availability

The raw data of the ten datasets are available at: http://www.csie.ntu.edu.tw/~cjlin/libsvmtools/.

## Supplemental Information

Supplemental information for this article can be found online at http://dx.doi.org/10.7717/peerj-cs.799#supplemental-information.

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
