# Peer review of "DPWSS: differentially private working set selection for training support vector machines"

_PeerJ Computer Science, doi:10.7717/peerj-cs.799_

## Round 0.1 · original submission · Major Revisions

Dear Authors,

Based on the suggestions from reviewers and my own observations I suggest major revisions for this paper. I would also suggest the authors to check this recent work: An adaptive multi-layer botnet detection technique using machine learning classifiers.

·

Basic reporting

Typos and English errors

Experimental design

no comment

Validity of the findings

no comment

Additional comments

What is the base of author claiming that DPWSS my be the first private working set selection algorithm based on differential privacy?
Abstract is well-written but can include more info about experimental results.
Related work section is too short. Authors should include some recent works and list their limitations.
Authors can include references such as:
A Review of Machine Learning Algorithms for Cloud Computing Security
Performance Assessment of Supervised Classifiers for Designing Intrusion Detection Systems: A Comprehensive Review and Recommendations for Future Research
Author Classification using Transfer Learning and Predicting Stars in Co-Author Networks

Unnecessary background has increased the paper length.
Can you compare your approach with an existing approach.
Pl revise conclusion to conclude the work.
Pl proof-read the manuscript.

Reviewer 2 ·

Basic reporting

Authors propose a new differentially private working set
selection algorithm (DPWSS), which utilizes exponential mechanism to privately select
working sets. To the best of our knowledge, DPWSS is the first private working set
selection algorithm based on differential privacy. The authors theoretically prove that the proposed
algorithm satisfies differential privacy. The extended experiments show that DPWSS can
achieve classification accuracy and optimized objective value almost the same as the
original non-privacy SVM algorithm with higher execution efficiency.

Experimental design

Authors propose a new differentially private working set
selection algorithm (DPWSS), which utilizes exponential mechanism to privately select
working sets. To the best of our knowledge, DPWSS is the first private working set
selection algorithm based on differential privacy. The authors theoretically prove that the proposed
algorithm satisfies differential privacy. The extended experiments show that DPWSS can
achieve classification accuracy and optimized objective value almost the same as the
original non-privacy SVM algorithm with higher execution efficiency.

Validity of the findings

Authors propose a new differentially private working set
selection algorithm (DPWSS), which utilizes exponential mechanism to privately select
working sets. To the best of our knowledge, DPWSS is the first private working set
selection algorithm based on differential privacy. The authors theoretically prove that the proposed
algorithm satisfies differential privacy. The extended experiments show that DPWSS can
achieve classification accuracy and optimized objective value almost the same as the
original non-privacy SVM algorithm with higher execution efficiency.

Additional comments

1. Please improve overall readability of the paper.
2. The objectives of this paper need to be polished.
3. Introduction is poorly written.
4. Relevant literature review of latest similar research studies on the topic at hand must be
discussed.
5. Result section need to be polished.
6. There are some grammar and typo errors.
7. Improve the quality of figures
8. Define all the variables before using

The authors can cite the following refrences
1. A Novel PCA-Firefly based XGBoost classification model for Intrusion Detection in Networks using GPU
2. A Novel PCA-Firefly based XGBoost classification model for Intrusion Detection in Networks using GPU

Reviewer 3 ·

Basic reporting

In this paper, the authors proposed a new differentially private working set
selection algorithm (DPWSS), which utilizes an exponential mechanism to privately select
working sets. This paper is providing a good contribution to the state of the art.

Experimental design

The extended experiments show that DPWSS can achieve classification accuracy and optimized objective value almost the same as the original non-privacy SVM algorithm with higher execution efficiency.

Validity of the findings

To the best of our knowledge, DPWSS is the first private working set
selection algorithm based on differential privacy

Additional comments

Please improve the abstract. It should highlight the background as well and add some factual results that shows how much results have your approach achieved.
2. Introduction is too long. Please reduce it. It should be specific.
3. in introduction.... Support vector machine (SVM)...... quadratic programming (QP......... Sequential Minimal Optimization (SMO).... make it in a single format.
4. The quality of the figures can be improved more.
5. Result section is written badly. Explain results extensively and clearly explain the flow of results.
6. All equations, tables and figure should be cited in the text.
7. The author should add more details of the results and improve the mathematical models.
8. Future work and Challenges are the important things that lead the researchers to take the current state-of-the-art to further step. The authors should explain What open issues and challenges are needed to be addressed by the researchers in this field?
9. What are the evaluations used for the verification of results (like, roc curve)?
10. Abbreviations and Acronyms should be added only the first time. Then only acronyms should be used in the entire paper.
11. Few latest references can be added and explained a bit in the related work section. some are given below
12. Problem statement must be in the introduction.
13. Grammar check is required.


Fire detection method based on improved fruit fly optimization-based SVM
A Novel PCA-Firefly based XGBoost classification model for Intrusion Detection in Networks using GPU
Privacy protection algorithm for the internet of vehicles based on local differential privacy and game model
Smo dnn Spider monkey optimization and deep neural network hybrid classifier model for intrusion detection
Frequent itemset mining of user’s multi-attribute under local differential privacy

---

## Round 0.2 · Minor Revisions

Dear Authors,
Even though the manuscript is significantly improved, it still needs some improvements before acceptance. The following comments as suggested by the reviewers are to be addressed..

1.Related work section is still too short. The authors didn't address the comment.
2. Proof-reading is required.
3. The significance of Equations 10 and 11 should be described.
4. In section 4.5 add a paragraph for privacy analysis process.

Note: The authors need not cite the suggested references from reviewers if they are irrelevant. Mention in the response sheet that the references are irrelevant.

·

Basic reporting

Proof-reading is still required.

Experimental design

NA

Validity of the findings

NA

Additional comments

Related work section is still too short. Authors didn't address the comment.
Authors should include some recent works and list their limitations.
Authors can include references such as:
A Consolidated Decision Tree-based Intrusion Detection System for binary and multiclass imbalanced datasets
Performance Assessment of Supervised Classifiers for Designing Intrusion Detection Systems: A Comprehensive Review and Recommendations for Future Research

Conclusion can be improved by adding limitations of the proposed work.
Proof-reading is required.

Reviewer 2 ·

Basic reporting

The authors have addressed all of my comments, paper can be accepted in the current form. Thank you for giving me this opportunity.

Experimental design

Good

Validity of the findings

Good

Additional comments

The authors have addressed all of my comments, paper can be accepted in the current form. Thank you for giving me this opportunity.

Reviewer 3 ·

Basic reporting

no comment

Experimental design

no comment

Validity of the findings

no comment

Additional comments

authors have addressed most of the comments, but need to revised all comments suggested.
Equations 10 and 11 should be described the significance.
in section 4.5 privacy analysis one paragraph is needed to describe the process in text.
suggested related references are not cited yet.

---

## Round 0.3 · Major Revisions

The PeerJ Editorial Office have asked me to take over the handling of this paper, as they had some concerns about the prior reviews. I appreciate this is not ideal, however I decided to invite new reviewers to assess the manuscript and their comments are below.

The reviewers consider that the paper has value and can be publishable, but they also give a set of clear suggestions to improve the work, by expanding the analysis and improving a number of aspects throughout. Please address their comments.

Reviewer 4 ·

Basic reporting

In this paper, the authors tackled an interesting problem of selecting a training set for support vector machine while maintaining privacy of the data. The topic is worthy of investigation, but the following issues need to be thoroughly addressed before the manuscript could be considered for publication:
1. There are quite a number of vague statements across the manuscript. As an example, what do the authors mean by saying (in the abstract): “The errors of optimized objective value between the two algorithms were nearly less than two”? And then, “the DPWSS algorithm had a higher execution efficiency by comparing iterations on different datasets” – higher than what? Than the “original non-privacy SVM”?
2. Please avoid using bulked references, such as “Support vector machine (SVM)[1][2][3][4][5]” – it does not bring any useful to the reader. Please unfold such bulked references.
3. The authors should clearly state what do they mean by the “working set selection”.
4. Please avoid naming sections as “DP” etc (I suggest unfolding acronyms in the section/subsection headers).
5. What is the time and memory complexity of the proposed algorithm?
6. The authors sort of failed to contextualize their work within the state of the art of training SVMs. There have been a multitude of methods proposed so far, which could potentially impact maintaining privacy of the classifier. As an example, there are a number of training set selection methods around which could be used for selecting such training examples that could help maintain privacy. Are such methods possible to combine with the proposed approach? It would be useful to discuss that to elaborate a bit more comprehensive view of the problem of SVM training. For some references on training set selection for SVMs, the authors may want to check e.g., https://link.springer.com/article/10.1007/s10462-017-9611-1, https://link.springer.com/chapter/10.1007/11539087_71.
7. Although the English is acceptable, the manuscript would benefit from proofreading (there are several grammatical errors around).

Experimental design

1. The characteristics of the datasets used in the experimental study must be expanded. Specifically, what is the imbalance ratio within each dataset? Does it have any impact on the overall performance of the optimizer? Also, the number of datasets is fairly small – currently, it is common to test such SVM-related algorithms over 50+ datasets to fully understand their generalization abilities. To this end, I suggest including much more benchmarks in the experimental validation (especially given that only one dataset may be considered “large”).
2. Please present all quality metrics used in the manuscript in a formal (mathematical way) through providing their formulae.
3. It would be useful to at least discuss if the proposed method is suitable for multi-class classification using support vector machines (e.g., in the one-vs-all strategy).
4. The authors should present all relevant quality metrics: AUC, accuracy, precision, recall, F1 and MCC.
5. The authors should back up their conclusions with appropriate statistical testing (here, non-parametric statistical tests) to fully understand if the differences between different optimizers are significant in the statistical sense.

Validity of the findings

In my opinion, the authors should extend the experimental validation (as suggested in my previous comments), to provide clear and unbiased view on the generalization abilities of the proposed techniques.

Reviewer 5 ·

Basic reporting

This paper presents a differentially private working set selection to train support vector machine. They focus on decomposition in the sequential minimal optimization and perturb each iteration with exponential mechanism. The novelty is quite limited to this optimization and no empirical comparison to any existing differentially private SVM. The paper is clearly organized while some of the terms and notations are not consistently explained, such as constant-factor. I also have major concerns for the illustration of the results and evaluation design as discussed below.

Experimental design

I notice that the authors emphasize that their scheme is free from the influence of high-dimensional data on noise while their evaluation is mostly on dataset with at most a few hundred dimensions. The author should clarify the definition of “high-dimension” in the case of differential privacy adoption in SVM and address this claim by using a substantially high-dimensional dataset.

Validity of the findings

For Figure 2 to 9, it is very challenging to tell which scheme is superior as they look the same to me. I highly suggest the authors to start the y-axis at a higher value such as 0.6.

Similar issues are on Figure 10-12 given none of the combination gives an error over 2.5. The authors can scale up the value to address the performance difference.

Also it is confusing to use curve chart when plotting the results for different \epsilon and \delta combination as they do not represent any sort of performance trend. If the authors intend to demonstrate the impact of different parameters, they should fix on one and vary the other. Two graphs or matrix tables are more suitable for that purpose.

Additional comments

Minor:
Section 3.1: for training a SVM -> an
Section 5.3: under the circumstances most of the violating -> under the circumstances, most

---

## Round 0.4 · Major Revisions

The current revision is unfortunately not satisfactory in addressing the proposed revisions. I would advise the authors to consider these suggestions seriously, and submit a new revision if they consider that these can be addressed. I do agree with the reviewers that these are important concerns needing to be addressed, and unfortunately we would otherwise not be able to give further consideration to the submission if these revisions are not in place.

Reviewer 4 ·

Basic reporting

Unfortunately, although the authors tried to address my concerns, the most important ones still remain unresolved. Specifically, please refer to the following comments from my previous review: 2), 3), 7).

Experimental design

The authors should still work on the experimental validation of the proposed algorithm. Specifically, the authors should still expand their benchmarks, present more detailed experimental results captured using the metrics I mentioned in my previous review, and should present clear statistical evidence (backed up with appropriate statistical testing) behind the claims. I consider the following comments from my previous review still unresolved: 1), 4), 5).

Validity of the findings

See my previous comments

---

## Round 0.5 · accepted · Accept

Following the last round of revisions, the paper can now be accepted for publication in its present form.

Reviewer 4 ·

Basic reporting

I am happy to see that the authors have addressed my concerns, and the manuscript is in better shape now.

Experimental design

-

Validity of the findings

-